# Purity of lithium metal electrode and its impact on lithium stripping in solid-state batteries

Juri Becker [1] ✉, Timo Weintraut[1,2], Sebastian L. Benz[1], Till Fuchs[1], Christian Lerch[1], Pascal Becker[1], Janis K. Eckhardt [1], Anja Henß[1,2], Felix H. Richter [1] & Jürgen Janek [1] ✉

Recent studies emphasize that incorporating lithium metal electrodes can increase the energy density of next generation batteries. However, the production of lithium metal with high purity requires multi-stage purification steps due to its high reactivity. Furthermore, subsequent handling under inert conditions is required to prevent degradation. To circumvent handling of lithium metal and further improve energy density, researchers are exploring reservoir-free cells often referred to as "anode-free" cells. Reservoir-free cells are assembled without using lithium metal. Instead, lithium is electrodeposited at the interface between a current collector and a solid electrolyte from positive electrode materials during the first charge. Despite the potential of reservoir-free cells, there is limited understanding of the purity of electrodeposited lithium metal and how impurities might affect the electrochemical kinetics. This study examines first the purity of electrodeposited lithium at the steel|$Li_6PS_5Cl$ interface. Then, it shows how impurities in lithium electrodes affect stripping capacity when using commercial lithium metal foils with both $Li_6PS_5Cl$ and $Li_{6.25}Al_{0.25}La_3Zr_2O_{12}$ as solid electrolytes. By using time-of-flight secondary mass spectrometry and X-ray photoelectron spectrometry, we reveal that a lithium layer with high purity is electrodeposited at the negative electrode in reservoir-free cells and that common impurities in lithium metal (reservoir-type) electrodes like e.g. sodium negatively influence the accessible lithium capacity during discharge.

Growing demand for higher power and energy densities in batteries triggers extensive research toward solid-state batteries (SSBs)[1,2]. The solid electrolytes (SEs) used in SSBs provide higher shear modulus than liquid or gel polymer electrolytes, potentially suppressing dendrite formation and increasing safety[3–5]. Thereby, SEs could potentially enable the use of the lithium metal electrode. Lithium has a low redox potential of $E_H = -3.04\,V$ and a theoretical specific capacity of $q_{th} = 3861\,mAh\,g^{-1}$, which could substantially increase the energy density compared to current cells with graphite electrodes[6,7].

Conventionally, lithium metal electrodes are studied using lithium foils produced via molten salt electrolysis in industry[8–10]. These foils usually contain impurities in the 0.1% to 1% range, particularly sodium, which are challenging to remove during fabrication. Additionally, degradation layers form on the foils' surfaces even under practically

[1]Institute of Physical Chemistry and Center for Materials Research (ZfM), Justus-Liebig-University Giessen, Heinrich-Buff-Ring, Giessen, Germany. [2]Institute of Experimental Physics I (IPI), Justus-Liebig-University Giessen, Heinrich-Buff-Ring, Giessen, Germany. ✉e-mail: juri.becker@pc.jlug.de; juergen.janek@pc.jlug.de

relevant inert gas conditions, increasing the interfacial resistance to SEs[11,12]. This issue complicates the preparation of reversible lithium|SE interfaces, often necessitating mechanical removal of the degradation layers or the use of high temperatures and pressures during cell assembly.

To circumvent the processing and safety challenges associated with reactive lithium foils, reservoir-free cell concepts (RFCs) are emerging[13–16]. In RFCs, lithium is stored only in the discharged (lithiated) active material (i.e., the positive electrode), without excess lithium at the negative electrode. Instead, lithium metal is electrodeposited on the current collector (CC) of the negative electrode during the initial charge, which increases energy density and simplifies production by eliminating the need for inert gas storage of lithium and avoiding its degradation. Moreover, established SEs such as $Li_7La_3Zr_2O_{12}$ (LLZO) or $Li_6PS_5Cl$ (LPSCl), with lithium transference numbers near unity[17–19], may yield chemically pure electrodeposited lithium compared to industrially produced foils[18,19]. This electrochemical purification effect and its influence on cell properties has previously been widely overlooked in battery research. As is shown in the following, the use of lithium metal even with a small fraction of impurities can seriously affect the kinetics of cells with lithium metal electrodes—particularly in fundamental research, where lithium metal foils of different origin are widely used.

In fact, much of the recent research on lithium|SE interfaces relies on the use of lithium foils[6,20,21], often without considering and specifying the impact of lithium purity. In particular, the kinetics and morphological development of lithium metal electrodes is often studied in apparently symmetric cells with two foil electrodes. In certain full cell concepts, such as lithium-sulfur or lithium-air batteries, the positive electrode materials are in their delithiated state during assembly[22,23], even necessitating the use of pre-formed lithium metal (e.g., lithium foil) at the negative electrode. We believe that lithium purity is a crucial factor across various cell types that has been largely disregarded so far. In contrast, in practical applications of RFCs, all cyclable lithium would initially be contained in the positive electrode materials, i.e., no lithium foils are used during assembly. After the initial charging step (i.e., after lithium metal electrodeposition) the high purity of the electrodeposited lithium layer likely affects continuous cycling of the cell much less. In practice, SSBs are often investigated with a minimal lithium reservoir on the negative electrode, aiming for the compensation of irreversible lithium loss. While these cells are still promising to achieve higher energy densities[24], the role of the impurity content in the lithium reservoir is not known.

Research on positive electrode active materials reveals that contaminants within these materials can significantly influence their electrochemical properties[25–27]. Yet, little is known about the impact of lithium metal purity on its electrochemical performance in SSBs[24]. Properties of metals in general vastly depend on their purity[28]. This implies that lithium metal electrodes may also exhibit variations in chemo-mechanical and microstructural properties depending on their purity[29–32]. As a result, the electrochemical properties, e.g., discharge capacity, of highly pure electrodeposited lithium in RFCs may also differ significantly from that of lithium foils produced industrially, which often contain impurities[30,32]. While some studies have addressed the filtering of impurities using solid electrolytes[19,33,34], the implications for RFCs remain unexplored, but are crucial for practical applications[28,32].

In this work, we investigate the electrochemical "filtering" of lithium metal by LPSCl using a steel|LPSCl|Li|steel model-type RFC. Additionally, we examine the influence of impurities on the accessible lithium inventory (i.e., stripping capacity) in Li|LPSCl|Li and Li|LLZO|Li model-type symmetric cells using lithium foils. The purity of electrodeposited lithium and the concentration of impurities at the SE interface with a commercial lithium electrode after cell operation are analyzed using time-of-flight secondary ion mass spectrometry (ToF-

SIMS) and X-ray photoelectron spectroscopy (XPS). Our analysis confirms that electrodeposited lithium layers exhibit high purity. In contrast, when lithium foils containing impurities are used as the lithium reservoir, these impurities become more concentrated during the lithium dissolution process. They accumulate at the SE interfaces and cause kinetic hindrance. We note that diffusion of impurities in lithium metal, like e.g. Na, has already been studied in the 1970s, which we will consider in the discussion of our results. Ultimately, our results reveal a correlation between lithium purity and accessible stripping capacity using solid electrolytes. We identify lithium purity as a key factor to improve the electrochemical performance of lithium metal electrodes in SSBs.

## Results and Discussion

### Electrochemical deposition of lithium at the working electrode

To investigate the purity of electrodeposited lithium, a lithium layer with a thickness of at least about 30 μm is needed. Here, a steel|LPSCl|$Li_{LP}$|steel model system was used for electrodeposition, due to its suitability for electrodepositing thick lithium layers[29,35,36]. The used cell setup and a characteristic voltage profile for electrodeposition of lithium are shown in Fig. 1.

The voltage profile confirms successful nucleation and electrodeposition of lithium at the steel|LPSCl interface. At the beginning of the measurement, a characteristic voltage minimum is observed, representing the nucleation overvoltage of $Li_D$ at the WE when referenced to the following plateau[13]. The constant plateau afterwards demonstrates continuous lithium electrodeposition at the WE and stripping at the CE. After ~12 mAh cm$^{-2}$ of transferred Li, an increase in overvoltage is observed due to a contact loss at the CE despite application of external pressure, which indicates depletion of the $Li_{LP}$ reservoir[37]. With this procedure, $Li_D$ films with a mean thickness of about 60 μm were electrodeposited at the WE, which were then examined for their purity in the following section. Due to lithium depletion at the CE and the resulting poor contact, the surface of the solid electrolyte was also accessible for surface analysis after being in contact with the lithium CE during stripping.

### Chemical analysis of electrodeposited lithium and reference samples

The three different lithium samples $Li_{LP}$, $Li_{HP}$ and $Li_D$ were then investigated regarding their purity using ToF-SIMS and XPS. Analyses of $Li_{LP}$ and $Li_{HP}$ were carried out on lithium foils produced from two

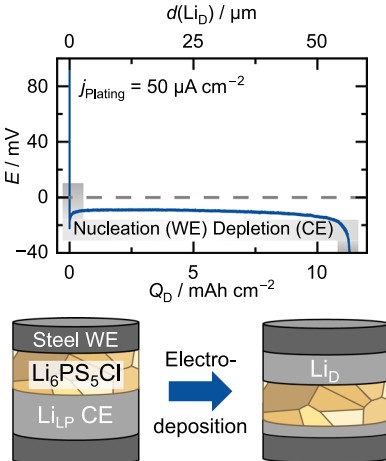

**Fig. 1 | Electrodeposition of a lithium layer.** Voltage profile during lithium electrodeposition at the steel|LPSCl interface using a current density of 50 μA cm$^{-2}$ and schematic of the used cell setup steel|LPSCl|$Li_{LP}$|steel before and after the lithium deposition. Herein $Li_{LP}$ is a lithium foil with low purity and $Li_D$ the electrodeposited lithium.

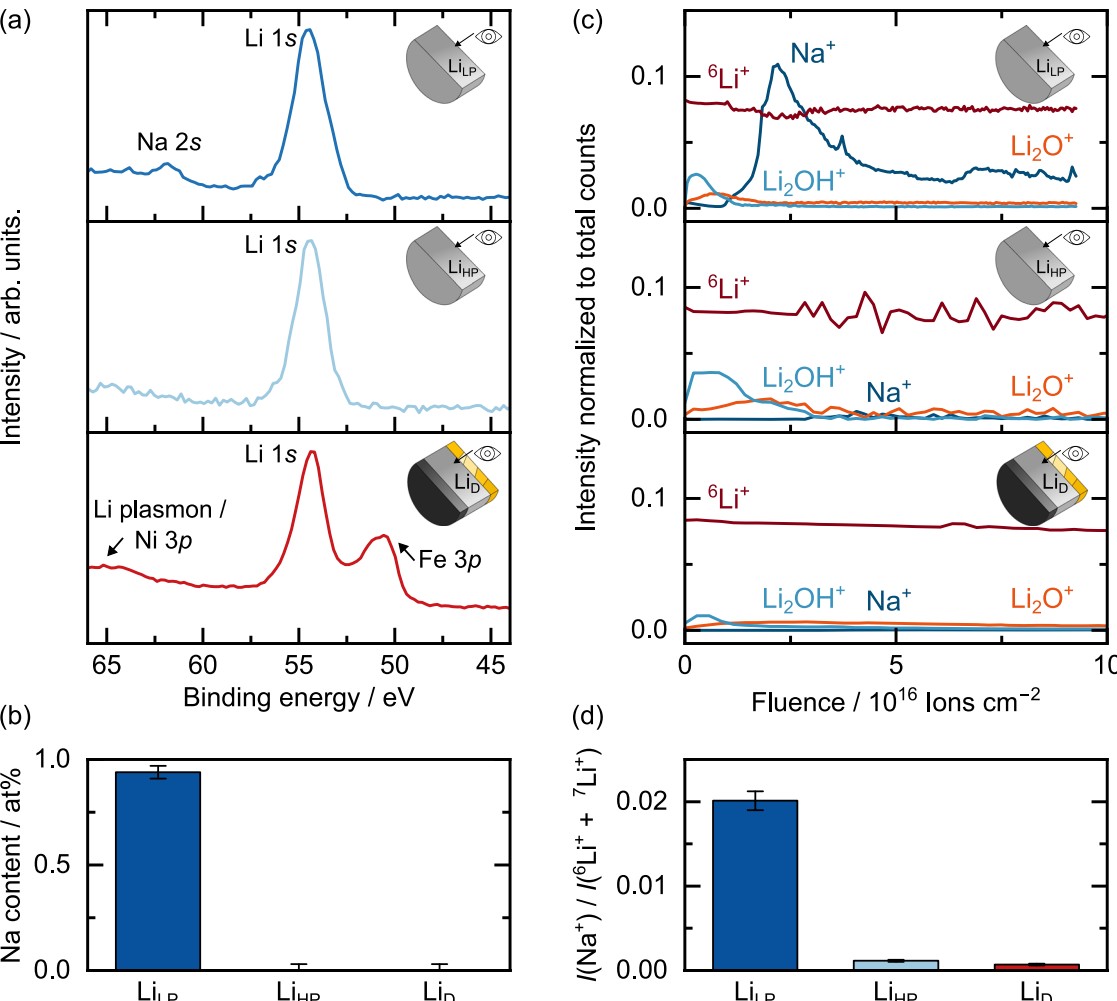

**Fig. 2 | Measurement of the impurity content of various lithium samples.**
**a** Acquired XPS spectra of $Li_{LP}$, $Li_{HP}$ and electrodeposited $Li_D$. **b** Calculated sodium content from the XPS analysis. In case of $Li_{HP}$ and $Li_D$, sodium impurities are below the detection limit, which is why only error bars are given. The error is ± 0.03 at% in all measurements, which is the detection limit of sodium within a lithium matrix using Al Kα radiation based on the work of A. Shard[42]. **c** ToF-SIMS depth profiles of $Li_{LP}$, $Li_{HP}$ and electrodeposited $Li_D$. The intensity of the signals was normalized to

the total counts. **d** Intensity ratios of the $Na^+$ and $Li^+$ signals (intensity $I$) from SIMS analysis. Data are presented as mean values from three measurements. The error bars correspond to standard deviation. All samples were measured in a cross-sectional geometry, as depicted schematically. $Li_D$ was analyzed ex-situ after electrodeposition of 12 mAh cm$^{-2}$ lithium at a current density of 50 µA cm$^{-2}$ in a reservoir-free cell system on a steel electrode (see Fig. 1).

different commercial lithium rods and serve as a reference to the electrochemically deposited lithium $Li_D$ (see Experimental section). According to the manufacturer's specifications, $Li_{LP}$ is lithium with low purity (99%), while $Li_{HP}$ has a higher purity (99.8%). To analyze $Li_D$, the electrodeposited layers (as shown in Fig. 1) were used. To avoid interference of surface passivation layers and LPSCl remnants, every sample was ion polished in a similar way and analyzed in a cross-sectional geometry. Although lithium contains various impurity species, including potassium, calcium, and magnesium, sodium was used as the reference signal for impurities in this study, as it is the most common impurity in lithium[38,39]. The acquired XPS spectra, ToF-SIMS depth profiles and the resulting quantification from these analyses are depicted in Fig. 2.

The comparison of the acquired XPS spectra for the different lithium samples in Fig. 2a already hints toward a decreased sodium impurity level in $Li_{HP}$ and $Li_D$ when compared to $Li_{LP}$. While sodium is detected in $Li_{LP}$ using the Na 2s signal, this signal is no longer visible for $Li_{HP}$ and $Li_D$. For $Li_D$, additional signals for Fe 3p and Ni 3p are observed. This artifact originates from the steel foil used as working electrode (i.e., as current collector for lithium electrodeposition) and the sample

holder used for the previous ion beam milling. For clarification, a similar sample after disassembly is shown in Figure S1.

For a quantitative comparison of the sodium content, the respective sodium concentrations in the lithium foils $Li_{LP}$, $Li_{HP}$ as well as the electrodeposited lithium $Li_D$ were calculated using the XPS spectra. Interestingly, it was found that the measured sodium content depends on the sputtering process. As shown as an example for the $Li_{LP}$ sample in Figure S2 and discussed in Supplementary Note 1, the intensity for the Na 2s signal increases with increasing sputtering time. After 360 s of sputtering with an acceleration voltage of 10 kV, the intensity approaches saturation. A subsequent increase of the acceleration voltage of the argon clusters to 20 kV significantly increases the Na 2s signal, indicating preferential sputtering or a mass related effect[40,41]. Furthermore, increasing the accelerating voltage while keeping the argon cluster size constant could lead to an increased removal of inorganic degradation products during argon cluster sputtering, as shown by Shard and Baker[41]. Sputtering at an acceleration voltage of 20 kV leads to an overestimation of the sodium content of around 5.4 at% in $Li_{LP}$. The surfaces of the samples were therefore sputtered using an acceleration voltage of 10 kV in between

measurements. The following quantification is based on the XPS spectra after sputtering with an acceleration voltage of 10 kV.

In $Li_{LP}$, a sodium content of around 0.94 at% is calculated, as can be seen in Fig. 2b. This value is in good agreement with the manufacturer's specifications for the sodium content of around 7000 ppm. In the case of $Li_{HP}$ the intensity of the Na 2s signal is too low to gain reliable values for the sodium concentration. Based on the work of Shard, the detection limit of sodium within a lithium matrix using Al $K_\alpha$ radiation is -0.03 at%[42]. This limit is therefore assumed as measurement error for the measurements performed here. The manufacturer's specification for the sodium content of $Li_{HP}$ is 820 ppm. Accordingly, sodium should be detectable. Deviations could stem from the measurement geometry, the surface sensitivity of XPS or the sputtering step. Nevertheless, the assumed measurement error of 0.03 at% is in the same order of magnitude as the manufacturer's specifications and is therefore used.

In the case of $Li_D$, no value for the sodium content can be specified based on the recorded XPS spectra either. Analogous to $Li_{HP}$, an error of 0.03 at% is assumed here. Clearly, $Li_D$ has a higher purity than its counterpart $Li_{LP}$, which served as the lithium reservoir at the counter electrode for lithium deposition at the steel CC. A reduction of the sodium content from 0.94 at% to 0.03 at% corresponds to a factor of 31. The purification of lithium by the solid electrolyte LPSCl, by only allowing lithium ions to migrate through its crystal structure, is therefore evident.

To quantitatively compare the sodium impurity levels, the Na 1s signal can also be used instead of the Na 2s signal. However, the binding energy of the Na 1s signal at $E_B = 1071$ eV is roughly twenty times higher than that of the Li 1s signal at 55 eV. At a constant excitation energy of -1487 eV (Al $K_\alpha$ source), the excited photoelectrons exhibit widely different kinetic energies. Accordingly, different penetration depths of the sample need to be considered for each signal. As shown in Figure S3, using the Na 1s signal results in a sodium content of 5.19 at% for $Li_{LP}$ and 0.74 at% for $Li_{HP}$. For $Li_D$, no significant intensity for the Na 1s signal was found. Apparently, the sodium content is overestimated using the Na 1s signal. However, qualitatively, our results indicate that $Li_D$ has an even higher purity than $Li_{HP}$.

The lithium cross-sections were further analyzed using ToF-SIMS, capable of detecting impurities at ppm to ppb levels, potentially distinguishing impurity contents between $Li_{HP}$ and $Li_D$. Figure 2c presents the respective depth profiles. As reference, the $^6Li^+$ intensity is displayed as it is in the same order of magnitude as the intensity of the signals of interest, i.e., as $Na^+$, $Li_2OH^+$ and $Li_2O^+$. Please note that, based on our measurements, the intensity ratio $I(^6Li^+) / I(^7Li^+)$ ratio is -0.09. Despite ion beam polishing under vacuum and inert sample transfer conditions, a surface passivation layer is visible on all samples, with an increased intensity of $Li_2OH^+$ and $Li_2O^+$. Notably, $Li_{LP}$ shows a sodium-rich layer beneath this oxygen-rich layer. We note that sodium segregation out of lithium metal has early been reported by Schily et al. and Powell et al.[43,44]. This layered structure could also be the reason for the strong increase in the Na 2s signal found following the sputtering step with 20 kV during the XPS analysis, in which more inorganic species such as LiOH and $Li_2O$ are sputtered, exposing a sodium rich layer underneath[41]. In contrast, $Li_{HP}$ and $Li_D$ show no significant $Na^+$ intensity, already indicating a reduced sodium content. To avoid an influence of the passivation layer, quantification was performed on all samples after cleaning by sputtering.

Since ToF-SIMS analysis provides only semi-quantitative data, the intensity ratios $I(Na^+) / I(^6Li^+ + ^7Li^+)$ presented in Fig. 2d should only be seen as a qualitative comparison. The error bars reflect the average of three measurements taken from different sample locations. For the lithium foil used as the counter electrode ($Li_{LP}$), the ratio $I(Na^+) / I(^6Li^+ + ^7Li^+)$ is -0.02. In the case of $Li_{HP}$ the intensity ratio is 0.001. For the electrodeposited lithium layer ($Li_D$), this intensity ratio is the lowest at around 0.0007. It is important to note the high sensitivity of ToF-SIMS,

particularly for sodium detection. Therefore, it is not surprising that sodium contamination is also detected in electrodeposited lithium. This could result from redeposition during ion beam milling or sodium diffusion from the solid electrolyte into the electrodeposited lithium. Despite the semi-quantitative nature of this analysis, the data clearly shows a significant reduction in sodium content from $Li_{LP}$ to $Li_D$, with the intensity ratio $I(Na^+) / I(^6Li^+ + ^7Li^+)$ decreasing by a factor of -29. This confirms the purification by LPSCl, as already indicated during XPS analysis. Additionally, the result obtained from the XPS analysis of the Na 1s signals, indicating that the sodium content in $Li_D$ is lower than in $Li_{HP}$, appears to be confirmed by the SIMS analysis.

The results are in good agreement with the reported high ion selectivity of solid electrolytes and was to be expected when considering the 60% difference in ionic radii for lithium and sodium. In LPSCl, lithium ions are tetrahedrally coordinated, have a single positive charge and an ionic radius of 0.59 Å[45]. For the same coordination number and charge, the ionic radius for sodium ions is 0.99 Å and is therefore substantially larger[46], making a hopping transport of sodium ions through the solid electrolyte with its tetrahedral regular lattice sites energetically unfavorable and thus unlikely.

The two lithium samples, $Li_{LP}$ and $Li_{HP}$ were additionally analyzed using EDX to obtain information about the distribution of sodium within the lithium foil. The lithium samples were attached to an LPSCl pellet, a capacity of 5 mAh $cm^{-2}$ was stripped from one of the two lithium electrodes, and a crater was milled in the stripping electrode by using a focused ion beam. The EDX analysis reveals sodium precipitates within the lithium matrix of $Li_{LP}$ (see Figure S4). In contrast, no such precipitates and almost no signal for sodium in general were found for $Li_{HP}$ (see Figure S5). These precipitates likely stem from lithium metal production from the melt, in which sodium impurities are inevitably present. Based on the binary phase diagram, lithium and sodium form a monotectic at low sodium content (see Figure S6)[47]. As lithium shows virtually no or minor solubility for sodium, the formation of sodium precipitates is to be expected, which is further discussed in Supplementary Note 2.

In summary, we conclude that only lithium ions can migrate through LPSCl, which is why a highly pure lithium layer with a sodium content below 0.1 at% is electrodeposited at the WE, even though roughly 1 at% of sodium is present in the lithium used as CE. Other impurities such as potassium, magnesium or calcium should also not be able to migrate through LPSCl, which is why they are also filtered out by electrodeposition[19,33,34]. When using reservoir-free cells, a high-purity lithium layer is electrodeposited onto the WE during the initial formation step. This purer layer is likely to differ from conventional lithium foils in terms of mechanical properties, such as hardness, and electrochemical properties, such as stripping capacities. The latter is discussed in the final chapter of this work for further analysis.

## Impurity analysis of lithium after lithium dissolution

Since impurities cannot migrate through the solid electrolyte, they must accumulate at the stripped CE. To investigate this, the surface of the solid electrolyte that has been in contact with an impurity containing lithium foil ($Li_{LP}$), acting as CE / lithium source for electrodeposition at the WE (see Fig. 1), is analyzed. Almost full lithium depletion allowed to peel off the $Li_{LP}$ electrode[37], enabling analysis of the LPSCl pellet surface using XPS and ToF-SIMS. The acquired XPS spectra and ToF-SIMS depth profiles, along with a reference measurement of a pristine LPSCl surface, are given in Fig. 3.

A comparison of the XPS spectra for the reference sample (LPSCl surface without lithium contact, shown in blue) and the LPSCl surface after being in contact with a $Li_{LP}$ counter electrode shown in red, reveals that a Na 2s peak is present only on the LPSCl surface that was in contact with $Li_{LP}$. This shows that an increased sodium concentration is present on the LPSCl surface after lithium stripping with an impure lithium foil. We attribute this to an accumulation of the sodium

impurities on the LPSCl surface as they are not able to migrate through the SE.

This finding is further supported by ToF-SIMS depth profiles. In the reference sample, the $^6Li^+$ signal remains consistently stronger than the $Na^+$ signal throughout the measurement. However, for the LPSCl surface that was in contact with a $Li_{LP}$ counter electrode, the $Na^+$ signal becomes more intense than the $^6Li^+$ signal, confirming the formation of a sodium-rich layer on the solid electrolyte surface. Interestingly, also an enrichment of $K^+$ can be seen on top of this $Na^+$ rich layer. Notably, a sodium signal is detected in both measurements, suggesting the presence of sodium impurities within the solid electrolyte itself, potentially originating from precursor contamination.

These findings are in good agreement with the precipitates found inside the $Li_{LP}$ sample using EDX (revisit Figure S4). As these precipitates are present within the lithium foil, they are consequently also

present at the interface during preparation. Furthermore, as the volume of the lithium electrode decreases during the stripping process, more and more of these precipitates approach the solid electrolyte surface.

To validate the accumulation of impurities at the solid electrolyte interface, additional experiments at an $Li_{LP}|LLZO$ interface were conducted. LLZO is studied as a potential separator SE for SSBs. It is practically stable in contact with lithium metal, allowing us to rule out degradation at the Li|LLZO interface in this case[48]. One electrode of a symmetrical $Li_{LP}|LLZO|Li_{LP}$ cell was electrochemically stripped until pores formed. This allowed the stripped $Li_{LP}$ electrode to be peeled off and the surface of the LLZO pellet to be analyzed[37]. Here, too, an enrichment of sodium on the surface of LLZO could be detected, which is even visible to the naked eye as a dark film on the SE. The results of this investigation using ToF-SIMS and XPS are summarized in Figure S7 and shortly discussed in Supplementary Note 3. In addition, EDX measurements were carried out at the edge of the electrode surface, which also show sodium accumulation for $Li_{LP}$ in the area of the previous electrode contact. The results of the EDX analysis can be seen in Figure S8.

These results suggest that impurities such as sodium accumulate on SE surfaces during electrochemical lithium stripping. As only lithium ions migrate through the electrolyte, impurities concentrate on the CE side, leading to the observed buildup. The impact of these impurities on electrochemical performance is yet unclear and is discussed in the next section.

### Influence of impurities on the discharge process of lithium electrodes

To investigate the influence of impurities on the electrochemical properties of lithium electrodes, the three analyzed lithium samples with different impurity contents (Na content in $Li_{LP} > Li_{HP} > Li_D$) are compared during electrochemical stripping in Fig. 4. Please note that all subsequent stripping experiments were carried out using galvanostatic electrochemical impedance spectroscopy with a current density of $100\,\mu A\,cm^{-2}$ with no external pressure applied.

To compare stripping capacities, symmetrical $Li_{LP}|SE|Li_{LP}$ and $Li_{HP}|SE|Li_{HP}$ cells were built. Both LPSCl and LLZO were used as SE. Lithium was then stripped until pore formation occurred, as evident by a strong voltage increase to 1 V. To investigate the stripping performance of $Li_D$, a lithium layer with a capacity of $5\,mAh\,cm^{-2}$ was initially electrodeposited at a steel|LPSCl interface with a current density of $100\,\mu A\,cm^{-2}$ at an external pressure of 15 MPa. The voltage profile during electrodeposition is displayed in Figure S9. Subsequently, the external pressure was released, the current direction was reversed, and lithium was stripped until a pore formation occurred. The voltage profiles during stripping until pore formation are compared in Fig. 4.

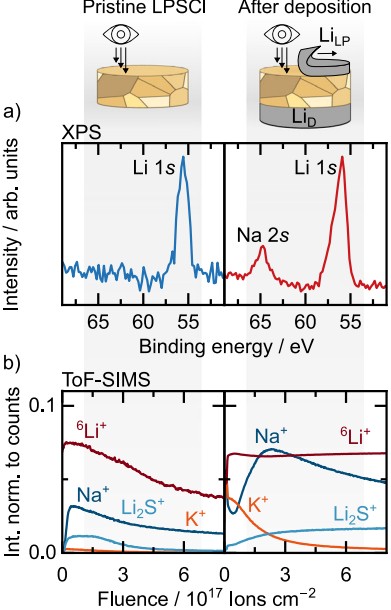

**Fig. 3 | Analysis of residues on the solid electrolyte surface after lithium stripping. a** XPS spectra of a pristine LPSCl surface on the left (in blue) and of an LPSCl surface that was in contact with $Li_{LP}$ CE during stripping on the right (in red). **b** ToF-SIMS depth profiles of a pristine LPSCl surface on the left and an LPSCl surface that was in contact with $Li_{LP}$ CE during stripping on the right. The intensity of the signals was normalized to the total counts. The LPSCl surface after deposition was analyzed ex-situ after stripping $12\,mAh\,cm^{-2}$ of $Li_{LP}$ in a reservoir-free cell (see Fig. 1).

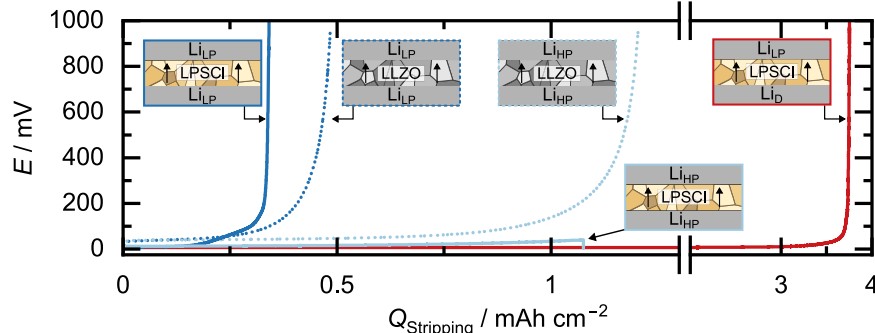

**Fig. 4 | The influence of the impurity content on the accesible lithium capacity during stripping.** Voltage profiles during stripping of lithium until pore formation in symmetrical Li|SE|Li cells. Cells using LLZO as SE are depicted with dotted lines. Cells using LPSCl as SE are depicted with solid lines. For clarification, the respective cell system is schematically depicted next to the respective data.

At this point, we note that varying impurity concentrations may alter the mechanical properties of lithium, potentially affecting the preparation of the Li|SE interface. However, the impedance spectra of both samples are similar, suggesting comparable interfaces (see Figure S10), with only a slight increase in the overall resistance in the case of $Li_{HP}$|LPSCl. This is likely as 15 MPa pressure were applied during electrode preparation with LPSCl, which is significantly higher than the yield stress of lithium at 0.7 MPa[6]. As a result, minor differences in the mechanical properties of the lithium samples are likely overshadowed by the high preparation pressure. A similar conclusion can be drawn for Li|LLZO interfaces, which are prepared at an isostatic pressure of 400 MPa, ensuring optimal interfaces with minimal resistance in both cases (compare Figure S11)[37].

A comparison of the voltage profiles during stripping shows that when lithium with a low purity ($Li_{LP}$, see Fig. 2) is used for LPSCl and LLZO as SE, contact loss occurs after ~0.34 mAh cm$^{-2}$ and 0.48 mAh cm$^{-2}$, respectively. In contrast, when lithium with a higher purity ($Li_{HP}$, see Fig. 2) is used, 1.1 mAh cm$^{-2}$ is stripped for LPSCl, even though a slightly higher initial resistance was observed. Please note that the shown $Li_{HP}$|LPSCl|$Li_{HP}$ sample is short-circuited before reaching the cutoff potential. The reason for this is dendrite growth within the SE. In the case of $Li_{HP}$|LLZO|$Li_{HP}$, 1.2 mAh cm$^{-2}$ can be stripped before reaching the cutoff of 1 V, which is close to experiments carried out in previous studies[37]. In conclusion, for both SEs, a substantial increase in stripping capacity by a factor of about 3 is observed when lithium with a higher purity is used. To ensure the reproducibility of the results, a set of three cells was examined for each cell type. The additional cells using LPSCl as SE are shown in Figure S12, those using LLZO in Figure S13. For additional information, we refer the reader to Supplementary Note 4.

From this, we conclude that impurities commonly present in commercial lithium, such as Na and K, decrease the accessible capacity during stripping of lithium foils. We attribute this effect to the formation of sodium-rich precipitates within the lithium electrode, as shown in Figure S4, likely occurring during the fabrication process from the melt. During lithium stripping, the volume of the electrode decreases, causing an increase in the relative sodium content. Since sodium is almost completely insoluble in lithium (as illustrated in Figure S6), additional sodium-rich precipitates may form, most likely at the Li|SE interface. This behavior aligns with recent findings by Mann et al., who reported the formation of a 3D sodium scaffold during the stripping process from a monotectic lithium-sodium mixture with 3 at % Na prepared through cold work[49]. In contrast to our findings, the authors report improved lithium stripping kinetics which they attribute to the 3D structured Na interlayer formed during the first stripping step. A direct comparison with these results is difficult, as only garnet SE was used and most experiments were run at 60 °C. Similarly, Yoon et al. also investigated the influence of several at% of sodium in lithium during stripping. Under pressure, an increase in stripping capacity was observed with increasing sodium content, which the authors attributed to the greater plastic deformability of the sodium-lithium mixture[50]. During stripping, there was also an accumulation of sodium at the solid electrolyte interface observed. As in the case of Mann et al., the resulting sodium layer appears to avoid current focusing in the subsequent plating step, as it keeps contact between the electrode and the solid electrolyte. Additionally, Park et al. reported an improved lithium stripping by introducing a Na-K interlayer at the Li|LLZTO interface[51]. More work will be necessary to resolve this apparent contradiction. We suggest that small Na atom fractions, typical for a homogeneously dissolved impurity, will deteriorate the interface conformity, while a larger fraction of sodium, dispersed as a second phase, may form a thick interlayer.

Interestingly, when analyzing the lithium electrodes of the Li|LLZO|Li samples after pore formation, a difference in the pore geometry seems evident (see Figure S14). One reason for this is the increased stripping time in the case of lithium foil with higher purity ($Li_{HP}$). On the other hand, the impurities present could also lead to a changed pore geometry. This hypothesis is supported by the fact that sodium precipitates can be found in places on the lithium residues using EDX (see Figure S15).

Notably, the stripping capacity of 1.2 mAh cm$^{-2}$ for $Li_{HP}$ | LLZO|$Li_{HP}$ is comparable to the stripping capacity achieved by Krauskopf et al. in symmetrical Li|LLZO|Li of 0.9 mAh cm$^{-2}$, using lithium with a similar purity of 99.8%[52]. This supports the purity dependence found in this study and highlights that the stripping capacity is primarily limited by the lithium electrode, rather than the solid electrolyte.

In the case of $Li_D$, an even higher stripping capacity of 3.74 mAh cm$^{-2}$ is achieved, corresponding to 75% of the previously electrodeposited 5 mAh cm$^{-2}$ of lithium. However, this increase in stripping capacity cannot be exclusively attributed to a higher purity of $Li_D$, as other parameters also change. On the one hand, the $Li_D$|LPSCl interface contact could be more homogeneous due to the electrochemical deposition than is the case for the conventionally prepared Li|LPSCl interfaces. In addition, the actual contact area between $Li_D$ and LPSCl could be larger than the geometrically assumed one, since deposition within the SE could also have occurred during initial electrodeposition. This hypothesis is supported by a cross-section prepared on the $Li_D$|LPSCl interface of a similar sample, revealing dendritic structures, i.e. lithium that has been deposited inside the solid electrolyte (see Figure S16).

For a more in-depth electrochemical analysis of the stripping properties of the investigated lithium electrodes, the impedance spectra during stripping are analyzed using the distribution of relaxation times (DRT). From this, contour plots are created, which are shown in Fig. 5 for each of the samples shown previously in Fig. 4. Additionally, to clarify how the contour plot is created, Fig. 5a includes selected impedance spectra in the Nyquist representation and all recorded impedance spectra during stripping in the Bode representation using the phase angle. The resulting DRT data for the $Li_{HP}$ | LPSCl|$Li_{HP}$ sample are shown in Fig. 5b. During the continuous stripping of lithium, there is an increase in resistance, as evident from the growing semicircle in Nyquist representation with an apex frequency of 22 kHz. The corresponding data point is marked with a filled circle. This is complemented by a more negative phase angle in the Bode plot at the corresponding relaxation time ($\tau \approx 10^{-5}$ s). In the DRT representation (Fig. 5b), the increase in resistance results in an increasing maximum of the distribution function $\gamma(\tau)$. As the total resistance $R$ corresponds to the area below this curve, the increase in resistance can also be seen here. This increase in resistance is finally shown in the contour plot by the corresponding color coding.

An analysis of the contour plot $Li_{LP}$ | LPSCL|$Li_{LP}$ reveals a bulk transport impedance signal for LPSCl at $\tau \approx 10^{-7}$ s. Deconvolution of bulk and grain boundary (GB) contributions is not possible in the measured frequency range. For $Li_{LP}$, a broad interfacial signal ($\tau \approx 10^{-5} - 10^{-1}$ s) can be seen for LPSCl, which appears to be based on two different processes. Both signals are highlighted with black arrows and contribute to a similar extent to the resistance increase shortly before contact loss ($Q_{Stripping} \approx 0.3$ mAh cm$^{-2}$), as can be seen from the color coding. Additionally, the recorded impedance spectra during stripping, depicted in Nyquist representation in Figure S10, reveal two semicircles.

In the case of $Li_{LP}$|LLZO|$Li_{LP}$, three distinct processes can be separated. For high frequencies, i.e., low relaxation times ($\tau \approx 10^{-7}$ s), the impedance contribution corresponds to the LLZO bulk transport. The resistance contribution is higher than in the case of LPSCl due to the lower conductivity of LLZO compared to LPSCl. At higher relaxation times ($\tau \approx 10^{-5}$ s), the impedance from the grain boundaries of the polycrystalline LLZO can be seen. These two contributions remain constant throughout the measurement, which is consistent with previous measurements on Li|LLZO interfaces and shows that no lithium is

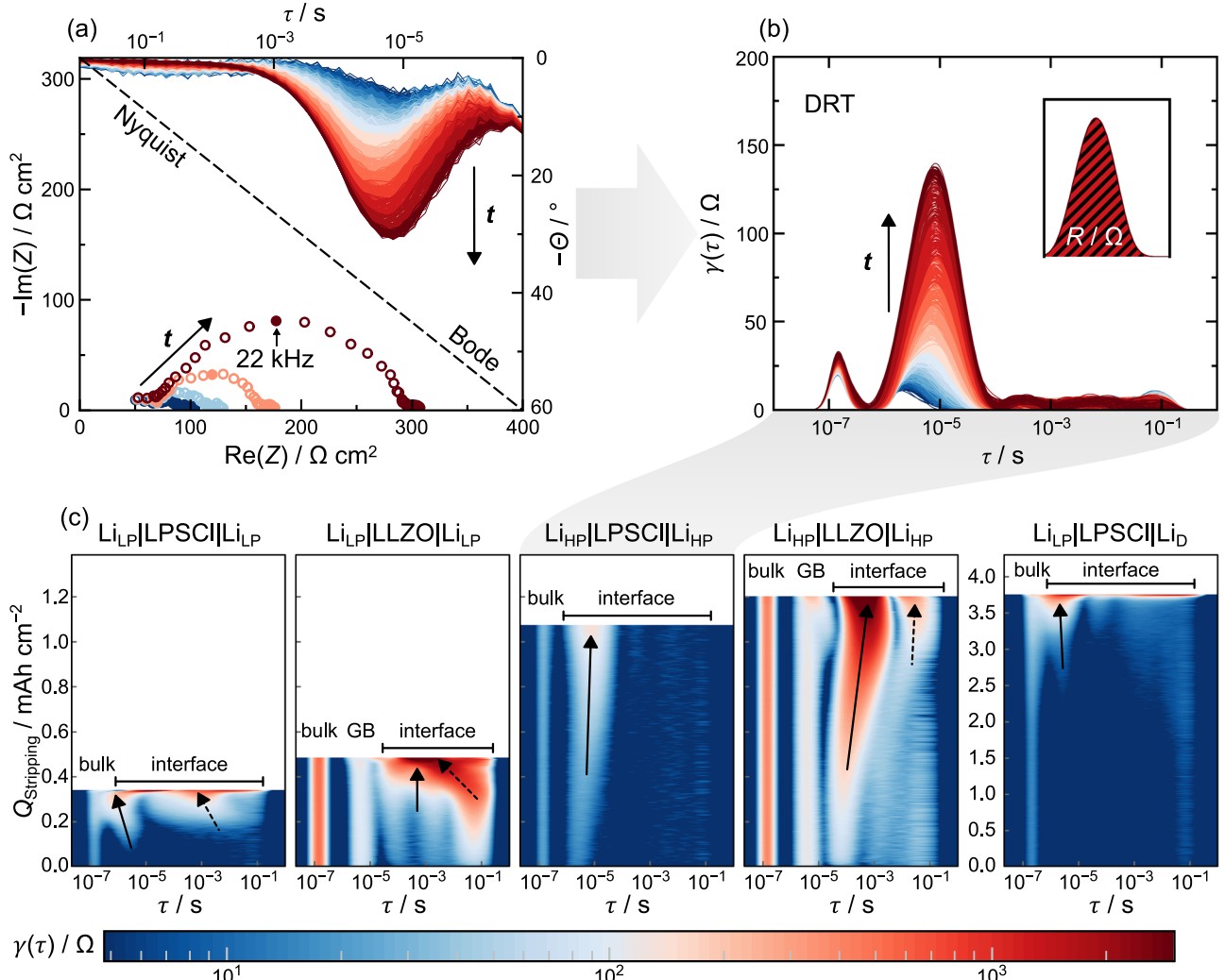

**Fig. 5 | Impedance analysis of the different lithium samples during stripping.** **a** Evolution of the impedance during stripping for the Li_HP|LPSCl|Li_HP sample showing all cycles in Bode representation (phase angle Θ versus relaxation time τ) in the upper part of the plot and selected cycles in Nyquist representation in the lower part of the plot. The characteristic frequency of the semicircle is marked with a filled circle. **b** Resulting distribution of relaxation time curves of the acquired data in (**a**). As shown in the small insert, the area below the function γ(τ) is the total resistance R for a process at a certain relaxation time τ. **c** Contour plots for the different samples, as marked with the heading, from Fig. 4 during stripping. The arrows highlight different interfacial contributions during stripping.

deposited inside the SE but is plated homogeneously on the CE instead[37,53,54]. Finally, at even higher relaxation times ($\tau \approx 10^{-4}$ s - $10^{-1}$ s), impedance contributions, which can be attributed to interfacial processes, become apparent. The signal occurring here is also very broad and two different processes, highlighted with two black arrows, are responsible for the increase in resistance during pore formation. This is visible in the impedance spectra in Nyquist representation (see Figure S11). Here, a broadening of the respective semicircle is evident.

For the middle contour plot (Li_HP|LPSCl|Li_HP), lithium with higher purity was used. As before, a constant bulk contribution is observed for LPSCl as SE. The impedance contribution of LPSCl is almost the same as for the Li_LP|LPSCl|Li_LP sample, highlighting the comparability between the two LPSCl pellets. However, when using Li_HP, only one single dominant interface signal is found at $\tau \approx 10^{-5}$ s. The signal is highlighted with an arrow inside the contour plot. As visible from the color code, the resistance increase during contact loss ($Q_{\text{Stripping}} \approx 1$ mAh cm$^{-2}$) is solely attributed to this signal. This also agrees with the shape of the impedance data in Nyquist representation (see Fig. 5a and Figure S10). Instead of having a broadening of the semicircle as in the case of Li_LP|LPSCl|Li_LP, only one semicircle is visible.

When using Li_HP in combination with LLZO, again, a constant bulk and GB contribution is observed. These impedance contributions of LLZO are almost the same as for the Li_LP|LLZO|Li_LP sample, highlighting the comparability between the two LLZO pellets. However, when using Li_HP, only a single dominant interface signal emerges between $\tau \approx 10^{-4}$ s - $10^{-1}$ s. Although there is a second signal at $\tau \approx 10^{-2}$ s (highlighted with a dashed arrow), which also becomes larger shortly before contact loss ($Q_{\text{Stripping}} \approx 1.2$ mAh cm$^{-2}$), the resistance contribution is at least one order of magnitude smaller at any time than for the signal at $\tau \approx 10^{-4}$ s - $10^{-3}$ s (highlighted with a straight arrow). This observation is again supported by the impedance spectra in Nyquist representation (see Figure S11), where almost no broadening of the respective semicircle can be seen.

For the last contour plot (Li_D|LPSCl|Li_LP), a slightly lower bulk contribution of the LPSCl pellet is visible in the beginning. During stripping and especially just before contact loss ($Q_{\text{Stripping}} \approx 3.8$ mAh cm$^{-2}$) this resistance contribution increases. As discussed above, we assume that during electrodeposition, lithium is also electrodeposited inside the LPSCl pellet, as shown for a similar sample in Figure S16. During the stripping shown here, this dendritic lithium is redissolved, decreasing the effective contact area between

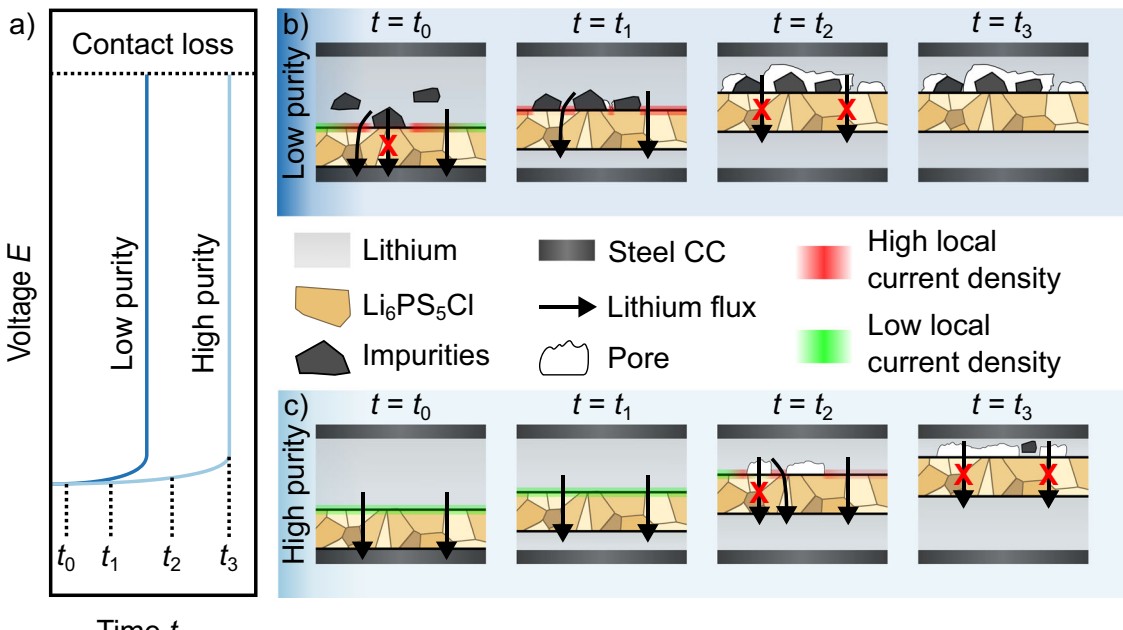

**Fig. 6 | Impact of lithium purity in (reservoir-free) solid-state batteries.** Scheme of the purification of lithium when using single-ion conducting SEs. **a** Voltage profile during lithium stripping for two different lithium samples, one with a high and one with a low purity. **b**, **c** Interface evolution between the two respective lithium samples and the SE during ongoing stripping. The different times are marked within the given voltage profile in (**a**).

lithium and LPSCl, thus increasing the resistance. However, the interface signal at $\tau \approx 10^{-5}$ is mainly responsible for an increase in resistance before contact is lost. This is consistent with the previous measurements using LPSCl. However, a second interface signal can also be seen for this sample at higher relaxation times, which only increases in intensity immediately before contact loss.

For both LPSCl and LLZO as SEs, the impedance evolution during stripping differs depending on the purity of the lithium used. With low-purity lithium, two (interface) processes contribute to the resistance increase during stripping, while high-purity lithium shows only one dominant interface signal. The impedance signal associated with pore formation during stripping—caused by kinetic limitations of the lithium electrode—occurs at $\tau \approx 10^{-4}$ s - $10^{-3}$ s for LLZO as SE. This is in good agreement with results from literature[37,52,54,55]. In the case of LPSCl used as SE, this relaxation time is around $\tau \approx 10^{-5}$ s, which also aligns with previous studies[56]. We hypothesize that the additional interface signals at higher relaxation times are due to impurities present at the lithium|SE interface. These impurities may stabilize additional pores with geometries different from those caused by kinetic limitations, as studies on constriction have shown that pore geometry strongly influences impedance signatures[57–60]. The impurities may also act as nucleation sites for pores[61]. Alternatively, the additional signal could be caused by the impurity particles themselves, which locally impede the lithium flux. These sodium-rich particles act as a locally blocking layer, resulting in an additional constriction contribution or even a completely blocking electrode layer[12,58,62].

**Impurities in reservoir-free model systems**
In the following, the observed purification by single-ion conducting solid electrolytes together with the influence of impurities during stripping of lithium is discussed. The schematics in Fig. 6 serve as a visualization of the drawn conclusions from the results above.

The given voltage profile shows stripping of two different lithium samples, one with a low and one with a high purity. The small offset on the time axis only serves for better visualization. As depicted in Fig. 6b and c, before start of electrodeposition ($t = t_0$), a homogeneous interfacial contact is given in both cases. However, in the case of impure lithium (Fig. 6b), impurities are already present at the interface with the SE.

With start of electrodeposition ($t_1 > t_0$), a lithium layer with high purity is electrodeposited at the working electrode for both samples, as only lithium ions can migrate through the solid electrolyte. The aforementioned impurities in case Fig. 6b lead to an increase of the local current density, as they block the local ion flux, decreasing the active electrode area.

For longer times of lithium stripping ($t_2 > t_1$), the lithium counter electrode with high purity Fig. 6c can still replenish the interface towards the SE, keeping contact. However, for the lithium electrode with impurities (Fig. 6b), voids already form at the interface. This is due to the accumulation of the contained impurities, increasing the local current density in their surroundings at remaining lithium contact spots and simultaneously blocking the local lithium flux where impurities are present, leading to a contact loss and an increase in the voltage as can be seen in Fig. 6a.

Prolonged electrochemical lithium dissolution ($t_3 > t_2$) eventually also leads to the formation of pores for the lithium electrode with high purity as shown in Fig. 6c. Reason for this is that the current density $j$ exceeds the rate of replenishment through vacancy diffusion at the Li|SE interface[37,63]. Thus, these vacancies accumulate at the interface and lead to the formation of pores, which ultimately leads to contact loss and an increase in the voltage as depicted in Fig. 6a. However, as indicated at $t = t_3$, the formation of impurity precipitates (i.e., sodium precipitates) might also occur in the case of lithium electrodes with high purity (Fig. 6c), as the lithium stripping increases the relative sodium content inside the electrode. According to the phase diagram, sodium precipitates should therefore form as lithium and sodium are not miscible. For the case of lower purity lithium (Fig. 6c), the interface does not change anymore, as pores were already formed at an earlier stage. The impurity enrichment likely also influences subsequent cycles. Studies of Na diffusion in Li by Mundy et al. suggest that Na may diffuse even faster than Li[44,64]. This would mean that any stripping of Li will cause a homogeneous increase of the Na concentration rather than a local increase at the stripped interface. Once stripping leads to Na supersaturation, precipitates will form and can decorate the stripped

interface. Thus, depending on the Na concentration in Li metal, Na precipitates may always occur and influence the electrode kinetics—but after different times.

These findings are in good agreement with theoretical works by Agier et al. on the void formation in metal electrodes with impurity particles in their interior, showing that impurity particles block the lithium flux at Li|SE interfaces[61]. The authors hypothesize that this leads to an inhomogeneous deformation of the lithium electrode, an increased dislocation density around the impurities and therefore to a preferential void formation in their surroundings. However, based on the calculations, these voids should not grow larger than 10 μm, taking micrometer sized impurity particles into account. Agier et al. further hypothesize that the observed electrode failure is caused by the coalescence of several smaller voids due to the dispersion of impurity particles within the electrode. During ongoing stripping of lithium, these accumulate at the Li|SE interface, stabilizing the formed voids.

This hypothesis aligns with our experimental findings in this study, in fact showing an accumulation of Na-rich particles at the LPSCl and the LLZO interface. We like to note that Agier et al. did chemically not specify the impurity they considered. In the case of $Li_{LP}$ this effect is more pronounced due to a higher content of impurities within the electrode, leading to a faster accumulation at the SE interface and a faster coalescence or formation of voids at the interface. For $Li_{HP}$, the impurity content is lower and therefore an accumulation of impurities at the interface leading to void formation is slowed down, if impurity driven at all. In the case of very pure lithium, contact loss may primarily result from the injection rate of vacancies at the Li|SE interface outpacing the replenishment rate of lithium to this interface. In a recent and apparently contradictory study mentioned above, Mann et al. intentionally prepared Li metal with a sodium fraction of 3 at%, which corresponds to the composition of the Li-rich monotectic. This fraction is significantly higher than in our case of $Li_{LP}$, and the authors observe a higher CCD than with their unmodified Li. After a complete stripping step of 21.6 mAh cm$^{-2}$ a Na network at the interface is found, that must have formed through the accumulation of Na precipitates. The authors assume that this network supports high current densities. We believe that the apparent differences compared with this work might stem from the manufacturing process used by Mann et al., likely also influencing the resulting lithium microstructure. As suggested by Yoon et al., differences in mechanical properties could also be the cause of the increased stripping capacities of lithium-sodium electrodes[50]. It remains to be clarified how the resulting sodium interlayer structure supports the lithium transfer. The mechanism might be similar as in the case of the Na-K interlayer introduced by Park et al.[51]. In our case, the typically low concentration of sodium in lithium clearly has a negative effect.

The impurity content likely influences the microstructure of different lithium samples[65]. Previous studies have shown that the microstructure of electrodeposited lithium differs substantially from that of lithium foils, with electrodeposited lithium forming large columnar grains[29,36]. This difference is likely accelerated by the high purity of electrodeposited lithium found in this study. Both the microstructure and the high purity of lithium are contributors to the observed differences in electrochemical behavior and may also influence the mechanical properties of lithium. Although we did not observe significant differences in interfacial resistances—potentially due to variations in Li|SE interfacial contacts resulting from different mechanical properties of the different lithium types—these differences might become more apparent under lower preparation pressures, approaching the yield strength of lithium of ~0.7 MPa. Notably, the mechanical differences are expected to be even more pronounced in electrodeposited lithium due to its even higher purity and columnar microstructure. Further research focusing on the mechanical properties of lithium, particularly in relation to its microstructure and purity, is necessary to gain deeper insights.

In this study, we examined the ability of the single-ion conducting solid electrolytes LPSCl and LLZO to filter impurities when using reservoir-free cells. Specifically, we analyzed the sodium content—being the most prominent impurity in lithium—in electrodeposited lithium layers and compared it to that in lithium foils. This comparison was conducted using complementary XPS and ToF-SIMS analyses. The results revealed a significant reduction in sodium content in the electrodeposited lithium layers, highlighting that solid electrolytes effectively filter impurities during lithium electrodeposition. Conversely, when using impure lithium foils as counter electrode or as lithium source for electrodeposition, the stripping process leads to an enrichment of impurities, i.e., sodium, in this electrode and at the SE interface, strongly reducing the stripping capacity of this electrode

By using two commercially available lithium foils with different impurity contents as well as an electrodeposited lithium layer, we found that impurities at the interface also affect the accessible lithium capacity before loss of contact. Herein, lithium foils with a higher purity showed a threefold higher stripping capacity. Accordingly, electrodeposited lithium layers will likely differ compared to commercially available lithium foils. This indicates that knowledge obtained from studying commercial lithium films may not be directly transferable to electrochemically deposited lithium films. This purity dependency needs to be considered in future studies, whether in model systems using lithium electrodes or in a possible application of lithium electrodes in lithium sulfur or lithium oxygen cell systems. Furthermore, the purity of the lithium used should always be specified when reporting cell data. We also like to note that true reservoir-free SSB cells will most probably be unaffected by any impurity accumulation at the solid electrolyte interface, provided that the positive electrode active material is sufficiently pure with respect to its lithium metal content. In fact, RFC may only operate well because of its inherent impurity filter effect. With additional deposition of dedicated metal interlayers, the electrodeposited lithium metal may be doped and modified such that improved electrode properties result. A better understanding of this will require in-depth studies of the impurity inventory of lithium metal before and after stripping and plating.

## Methods

### LLZO synthesis

LLZO was prepared using the reactants zirconium oxide ($ZrO_2$ < 100 nm, Sigma-Aldrich), aluminum oxide ($Al_2O_3$, 99.99%, ChemPur), lanthanum hydroxide ($La(OH)_3$, 99.9%, Sigma Aldrich) and lithium carbonate ($Li_2CO_3$, 99.998%, Puratronic). These were weighed with an excess of 3% $Li_2CO_3$ and homogenized using a ball mill (type: Fritsch Pulverisette 7 Premium Line, 10 Min @ 350 RPM, 20 Min break - 24 cycles) in a ball mill cup made of $ZrO_2$ inside and $ZrO_2$ balls as milling media. The intermediate product was pressed under air, using a steel die with a diameter of 25 mm, and calcined under an oxygen atmosphere at 1000 °C. The pellets were transferred to a glove box (argon atmosphere, with traces of $H_2O$, $O_2$ < 0.1 ppm and $N_2$ < 10 ppm and ground there. The intermediate product was examined for impurities using an X-ray diffractometer. A further grinding step (10 Min @ 350 RPM, 20 Min break - 40 cycles) was carried out in the ball mill. The ground powder was prepressed manually using a steel die with a diameter of 10 mm in the glove box and then isostatically pressed at 400 MPa inside sonodomes. The pellets were covered with mother powder and transferred to an oven where they were sintered at a temperature of 1230 °C for 1 h under an oxygen atmosphere[37]. The product was transferred back into the glovebox and the excess mother powder was removed and the pellets were polished for further processing, resulting in LLZO pellets with a diameter of ~8 mm and a thickness of ~2 mm.

### Cell preparation

Cell preparation was conducted inside a glovebox (MBraun, Germany) with traces of $H_2O$, $O_2$ < 0.1 ppm and $N_2$ < 10 ppm. For cell assembly, an

in-house fabricated casing was used. Steel stamps, which also serve as electronic contact, were used to press the cell components. A steel foil (thickness = 20 μm, AISI 304, Goodfellow, Germany), being the working electrode in the case of RFCs, was placed on top of one of the steel stamps. The $Li_6PS_5Cl$ powder (NEI Corporation, USA, particle size < 5 μm) was loaded onto the steel foil and compressed afterwards, using a second steel stamp. The cell was sealed using nitrile butyl rubber rings and compressed outside the glovebox, using a uniaxial press at 3 t for 1 min from each side. As a counter electrode, a freshly prepared lithium foil was used[37]. To remove surface passivation layers, a piece of lithium was cut from a lithium rod and hand pressed between two pouch foils, resulting in a lithium foil with a thickness of ~100 μm. As far as $Li_{LP}$ (LP = "low purity") is concerned, a lithium rod from Goodfellow (Lithium rod, purity 99%, Goodfellow GmbH, Germany) was used. In the case of $Li_{HP}$ (HP = "high purity"), a lithium rod from MaTecK (Lithium rod, purity 99.8%, Material, Technologie & Kristalle GmbH, Germany) was used. Analysis data from the manufacturers on the impurity content of the lithium rods can be seen in Table S1. One lithium electrode (d = 9 mm) was placed on one surface of the compressed LPSCl pellet for electrodeposition experiments with RFCs. For symmetrical Li|LPSCl|Li cells, a lithium electrode was placed on each side of the pellet. In all cases, an additional steel foil was placed between the steel stamp and the lithium electrode for easier cell disassembly. Finally, the cell was sealed, and an external pressure of 15 MPa was applied using an aluminum pressure frame. For electrodeposition within RFCs, electrochemical characterization was conducted within this pressure frame. For symmetrical cells, the pressure was released after an equilibration time of 12 h and lithium dissolution experiments were conducted without external pressure. Using LLZO as SE, lithium electrodes were prepared in the same manner as described above. After placing a lithium electrode (d = 6 mm) on each surface of the LLZO pellet, it was transferred into a pouch bag and vacuum sealed. The pouch was then isostatically pressed at a pressure of 450 MPa for 45 min. Once pressed, the symmetrical Li|LLZO|Li cell was removed from the pouch bag. The cell was then connected using nickel tabs, vacuum sealed again inside a new pouch and subsequently electrochemically characterized.

## Electrochemical characterization

Unless otherwise stated, all electrochemical experiments were carried out in a climate chamber at a temperature of 25 ± 0.1 °C. Initially, the cells were equilibrated for 6 h. Within this resting time, potentiostatic impedance spectra were recorded at an interval of 30 min using a potentiostat (VMP 300, Biologic, France) at open-circuit voltage conditions. The frequency range was set between 7 MHz and 100 mHz with 7 points per decade in logarithmic spacing at an amplitude of 10 mV. After equilibration, lithium was electrodeposited at the steel|LPSCl interface using chronopotentiometry (CP) at a current density of 50 μA cm⁻² in the case of RFCs for a capacity of roughly 12 mAh cm⁻². In the case of symmetrical Li|LPSCl|Li or Li|LLZO|Li cells, lithium was electrochemically deposited/dissolved, using galvanostatic electrochemical impedance spectroscopy (GEIS) at a current density of 100 μA cm⁻². The frequency range was set between 7 MHz and 1 Hz with 6 points per decade in logarithmic spacing at an amplitude of 10% of the direct current. To ensure reproducibility of the observed trends, a set of three cells were examined for each symmetrical cell system.

## Sample preparation

All preparation steps were carried out in the gloveboxes mentioned above. To examine $Li_{LP}$ and $Li_{HP}$ using XPS and ToF-SIMS, pieces were cut from the corresponding lithium rod and the passivation removed using the same procedure as described above. These pieces were then used to prepare cross-sections. To investigate $Li_D$, the cell used for lithium electrodeposition was disassembled. The working electrode was then removed from the LPSCl pellet using tweezers and

a cross-section of the entire working electrode was prepared. Please refer to the following paragraph for details on cross-section production. To investigate the LPSCl or LLZO surfaces after $Li_{LP}$ contact, cells were disassembled after a loss of contact between the $Li_{LP}$ electrode and the SE was caused by the stripping experiment. The electrode was removed using tweezers, exposing both the solid electrolyte surface and electrode surface. All transfer steps from glove boxes to any device were carried out using an inert transfer system (Leica VCT500, Leica Microsystems, Germany).

## Cryogenic ion-milling

Cross-sections of electrodeposited lithium at the working electrode or the $Li_{LP}$ and $Li_{HP}$ samples were prepared using an ion beam milling system (Leica EM TIC 3X, Leica Microsystems, Germany). Using liquid nitrogen, the temperature during milling was decreased to −120 °C. The device was operated at an accelerating voltage of 6 kV and an $Ar^+$ current of 2.2 mA. The prepared cross-sections were transferred to the SIMS or XPS systems using an inert transfer system (Leica VCT500, Leica Microsystems, Germany).

## Focused-ion-beam cutting and scanning electron microscopy

Cutting with focused ion beams and imaging were carried out with an XEIA3 GMU/plasma machine from TESCAN, which works with a Xe plasma.

## Scanning electron microscopy and energy-dispersive spectroscopy

Scanning electron microscopy was conducted within a Gemini SEM 560 from Carl Zeiss Microscopy GmbH. In case of energy-dispersive spectroscopy measurements an X-Max Extreme Silicon drift detector from Oxford Instruments was used.

## Secondary ion mass spectrometry

Elemental analysis was performed using an M6 Hybrid secondary ion mass spectrometer (IONTOF GmbH, Münster, Germany). After disassembly of the cell, depth profiling of the LPSCl pellet surface was performed in positive polarity using 30 keV $Bi^+$ primary ions at 80 μs cycle time with non-interlaced DSC-S 2 keV $Ar^+$ sputtering between analysis steps. The analysis area was set to 100 • 100 μm² with 128 • 128 pixels while the sputter area was set to 300 • 300 μm² (total dose density $8.45 • 10^{17}$ ions cm⁻²).

All lithium metal samples were first sputter cleaned in a 300 • 300 μm² area with 1 keV $Ar^+$ ions (total dose density $1 • 10^{17}$ ions cm⁻²) to remove the native oxidation layer formed on the lithium metal during transfer to the SIMS machine. Please note that the $^7Li^+$ intensity was corrected using the $^6Li^+$ intensity. Three surface spectra per sample were recorded on the cross-sections of the electrodeposited lithium (prepared by ion milling) in areas of 50 • 50 μm² with 128 • 128 pixels. For the lithium metal reference measurements, a surface spectrum of 200 • 200 μm² area and 128 • 128 pixels was recorded at three different, previously cleaned areas. In all cases, i.e., the electrodeposited Li and the reference Li samples, the $Na^+/Li^+$ ratio was calculated by dividing the $Na^+$ counts by the sum $^6Li^+ + ^7Li^+$ of counts. All data evaluation was conducted with SurfaceLab 7.4 software (IONTOF GmbH, Münster, Germany).

## X-ray photoelectron spectroscopy

Complementary to the SIMS investigations, the electrodeposited lithium as well as the lithium counter electrode was analyzed using a XPS machine of the type Phi Versaprobe IV. Monochromatic Al $K_\alpha$ radiation ($E_B = 1486.6$ eV) was used for excitation. Bulk samples were analyzed with an excitation power of 50 W resulting in an X-ray beam diameter of 200 μm, while electrodeposited samples were excited with a power of 5 W, yielding a 20 μm beam diameter. Lower excitation power was balanced by increasing the measurement time per point by

the reciprocal factor. The samples were mounted electronically insulated from the stage. A combination of low energetic $Ar^+$ ions and electrons was used for charge neutralization. Post referencing to the O 1 s signal ($E_B = 529.0\,eV$) ensured energy calibration. Survey spectra were recorded with a pass energy of 224 eV, while 55 eV were used for detail spectra. Alternating gas cluster ion bombardment was used to clean the surface in between spectra acquisition. The initial spectrum was recorded without sputtering beforehand. After acquisition, the surface was sputtered for 120 s and the second spectrum was acquired. This process was repeated multiple times. The acceleration voltage of the Ar clusters was either 10 kV or 20 kV. Data processing has been conducted with CasaXPS (Version 2.3.26rev1.1 P). The relative fractions $X_i$ of Li and Na were obtained respecting the chemical sensitivity (employing tabulated relative sensitivity factors $RSF_i$ of the XPS apparatus' manufacturer for orbital $i$), the transmission function of the apparatus $T(E)$ and the universal inelastic mean free path curve. A background correction has been applied using a Shirley-background.

## DRT analysis

The collected impedance data were analyzed in the time domain using the commercial software RelaxIS 3 (version 3.0.22.29, rhd instruments GmbH & Co. KG). Data in the high- and low-frequency ranges that showed residuals greater than 3% in the Kramers-Kronig test were excluded from the analysis. Only the real part of the impedance was considered, as the residuals were limited to just a few 0.1%. No additional data points were generated through interpolation, and the second derivative of the distribution function $\gamma(\tau)$ with a lambda parameter of $\lambda = 10^{-3}$ was applied in the Tikhonov regularization problem[66].

## Data availability

Source Data file has been deposited in "JLUPub" under accession code https://doi.org/10.22029/jlupub-19895.2[67].

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

## Acknowledgements

The authors want to thank Matilde Pavan for XPS measurements. The authors (JJ, JB) acknowledge funding by Evonik Operations GmbH. J.K.E. acknowledges financial support from the German Federal Ministry of Education and Research (BMBF) within the cluster of competence FESTBATT (project FB2-Thio: 03XP0430A) and by the Hessian State Ministry of Higher Education, Research, and the Arts (HMWK). This work has been partly funded by the German Federal Ministry of Education and Research (BMBF) within the DE-US program under the project "LiSI2", grant identifier 03XP0509B.

## Author contributions

J.B.: conceptualization, methodology, investigation, validation, formal analysis, writing–original draft, visualization, writing–review & editing; T.W.: methodology, investigation, validation, writing–original draft, visualization, writing–review & editing; S.L.B.: methodology, investigation, validation, writing–original draft, visualization, writing–review & editing; T.F.: conceptualization, methodology, writing–original draft, writing–review & editing; C.L.: investigation, validation; P.B.: investigation, validation; J.K.E.: formal analysis, validation, writing–original draft, visualization, writing–review & editing; A.H.: writing–original draft, writing–review & editing, supervision; F.H.R.: writing–original draft, writing–review & editing, supervision; J.J.: conceptualization, methodology, resources, supervision, writing–original draft, writing–review & editing, funding acquisition.

## Funding

## Competing interests

The authors declare no competing interests.
