## [Transparent Peer Review file · Nature Communications]

Purity of Lithium Metal Electrode and its Impact on Lithium Stripping in Solid-State Batteries

Corresponding Author: Professor Juergen Janek

Version 0:

Reviewer comments:

Reviewer #1

(Remarks to the Author)

The paper entitled "Purity of Electrodeposited Lithium and its Impact on Lithium Stripping in "Anode-Free" Solid-State Batteries" evaluates the purity of electrodeposited lithium at the interface of steel/Li6PS5Cl solid electrolyte in reservoir-free cell concepts and its influence on lithium stripping in symmetric cells using either Li6PS5Cl or Li6.25Al0.25La3Zr2O12 solid electrolytes. While the research findings are quite interesting and promising, more detailed explanations should be given for a better understanding of the manuscript. The following points need to be addressed prior to publication.

- In Figure 2a, Li 1s XPS spectra of LiLP, LiHP and LiD show different shapes. Particularly, Li 1s XPS data of LiHP contain 2 overlapping components and another small peak around 55 eV is detectable for LiLP, suggesting the existence of different Li-based species at the surface of LiLP and LiHP foils. This is expected to impact the interfacial compatibility between the lithium foils and solid electrolyte. The authors should deconvolute XPS core peak with the assignment of each component.
 - The authors should explain better the sputtering process for XPS analysis (paragraph starting line 235). In S2, how can the authors differentiate Li preferential sputtering and degradation? The apparition of a second Li 1s component at lower BE for 20kV needs to be explained.
 - For ToF-SIMS depth-profile, the authors should explain better the depth profiles especially for LiLP: why the sodium signal is increasing after the lithium one is decreasing? Does this mean the sodium is located underneath the lithium? The spatial distribution in depth of the different species should be more explained/developed.
- Moreover, it would be interesting to investigate the full positive and negative ToF-SIMS mass spectra to see if others species than Li⁺, Na⁺, Li2OH⁺ and Li2O⁺ can be of interest; indeed, the authors focused on Na impurities but what about the other species formed at the interfaces during the different processes.
- It is recommended to provide the XRD spectrum of the as-synthesized LLZO phase as well as a brief discussion regarding the phase purity.
 - The Li⁺ ionic conductivity of the prepared LLZO phase should also be provided.
 - Please add S to XPS in lines 220, 223, 228, 236, 245, 256, 326, 331, 335
 - The authors should organize better the different parts of the article by adding subtitles (ex: for EDX...)
 - The quality of the image in Figure S6 should be improved
 - The monotectic phase could be more explained/developed (it is not clear on the S6 figure)
 - Figure S8: the authors should mention that the intensity signal is not comparable between the two images otherwise the proportion of Na seems to be higher for LiHP
 - Line 388: it should be written figure S11 and not S10.
 - The paragraph from lines 427 to 435 is not clear and needs to be revised.
 - Line 532: it should be written Figure 6b and not 6c.

Reviewer #2

(Remarks to the Author)

Reviewer #3

(Remarks to the Author)

In this paper, purity of electrodeposited lithium and its impact on lithium stripping in “anode-free” solid-state batteries are discussed. Although the purity of lithium metal anodes is usually ignored in lithium metal batteries, the impact of impurities on electrochemical performance in anode-free systems still requires deeper consideration. Here are some questions that need to be clarified and answered by the authors.

1. The influence of cathode impurities on lithium stripping is mentioned in the abstract (it shows how impurities at the positive electrode affect stripping capacity.....) and conclusion (We also like to note that true reservoir-free SSB cells will most probably be unaffected by any impurity accumulation at the solid electrolyte interface.....). However, the manuscript does not provide any reference to the test results regarding the impact of cathode impurities on lithium deposition and stripping. In addition, the filtration capacity of solid-state electrolytes is also mentioned by the authors. Therefore, whether the impurities in the cathode will have an effect on the lithium metal anode requires more careful clarification by the authors.
2. The key scientific issues of the manuscript require the authors to consider. Through the experiments, the authors conclude that LiD has the highest purity. In the anode-free system selected by the authors, all lithium metal originates from LiD, which is virtually free of impurities. Therefore, the necessity of the topic of “purity of electrodeposited lithium and its impact on lithium stripping in anode-free solid-state batteries” needs further explanation.
3. Although the authors also state that minimal lithium reservoir is used to compensate for irreversible lithium losses, this impurity preferentially affects the lithium deposition. Therefore, the influence of impurities on lithium deposition needs to be discussed by the authors. And how does LiD affect the stripping behavior after it is deposited in lithium containing impurities?
4. The selected ranges of 1 and 2 in Figure S5 are not consistent with the shape of the local magnification figures, and the authors need to accurately select the magnification range.
5. In P10 the authors speculate that the impurity may originate from the diffusion of Na from the solid-state electrolyte to the electrodeposited lithium. Instead in P11 the authors emphasize “we conclude that only lithium ions can migrate through LPSCI”. These are contradictory results.
6. In the XPS analysis of Figure 3a, the authors analyzed the occurrence of sodium accumulation on the surface of the solid-state electrolyte. However, the peak ratio of Na 2s to Li 1s does not seem to differ significantly from that in LiLP (Figure 2a). How do the authors account for sodium enrichment? This needs to be clarified by the authors. Similarly, ToF-SIMS results (Figure 3b) are virtually indistinguishable from pure LiLP (Figure 2c). Therefore, the sodium-rich layer and sodium accumulation require more convincing evidences.
7. Why does the intensity of LiHP noise in Figure S8 even exceed that of Na in LiLP?
8. In Figure 4, the authors are advised to repeat the experiment several times to provide statistical results, thereby ensuring the accuracy of the experimental results.

Reviewer #4

(Remarks to the Author)

This study studied in detail the effect of commercially available lithium metal foils and "Anode-Free" lithium electrodeposition on the stripping capacity of the anode. If it can be put into commercial use, it will make it possible to create a rechargeable battery with a higher energy density than existing LIBs. To this end, it shows a new path from the perspective of the effect of impurities in the electrode, in terms of how to achieve reversible and stable charging and discharging of the Li metal anode. However, in the final discussion, it seems that a lot of information has been omitted in order to conceptualize the topic, and many questions remain. If possible, could a more detailed discussion be considered from both sides of the solid electrolytes Li6PS5Cl (LPSCI) and Li6.25Al0.25La3Zr2O12 (LLZO) that were discussed in this study? This is because, for example, the SEM analysis of the stripped Li electrode in Fig. S12 does not include a discussion of the differences between the solid electrolytes in Fig. 6. I feel that the impact is significant, so I would like you to consider this. In addition, the short circuit when using a lithium anode with LPSCI as the solid electrolyte is described in Fig. 4. This suggests the possibility that Li is penetrating the solid electrolyte. Do these correspond to the work of Porz (10.1002/aenm.201701003) or the different claims of Shen (10.1021/acscenergylett.8b00249) or Han (10.1038/s41560-018-0312-z), etc., which of the different claims do you correspond to? Is it difficult to consider it changes depending on the rigidity of the solid electrolyte? When these considerations are added, I feel that the discussion in Fig. 6 also expands.

In general, my comments do not deny the main point of the paper, but are a request for a little more in-depth analysis. I really like the approach of this study.

Reviewer #5

(Remarks to the Author)

This paper provides new insight into how to improve performance of all-solid-state batteries with lithium anode. The paper developed a new mechanism that impurity will accumulate at the lithium/solid electrolyte interface and deteriorate performance. The analysis is comprehensive. I have a couple of minor questions.

1. Could authors repeat the stripping experiment to show that the difference in stripping capacity is statistically significant?
2. Alloying lithium with other metal is often considered an effective strategy to improve the reversibility of lithium metal anode in all-solid-state batteries. So why is impurity not good, which can be considered as an extreme case of alloying? Is it because Na and Li do not alloy? I suggest authors discuss this point. For example, if Zn or In is intentionally doped inside as

impurity, will stripping capacity increase or not? This could help better understand this phenomenon.

Version 1:

Reviewer comments:

Reviewer #1

(Remarks to the Author)

The authors have taken all our remarks into account, responded correctly and precisely to our comments/questions and modified the manuscript accordingly; it therefore deserves to be published in this revised version.

Reviewer #2

(Remarks to the Author)

Reviewer #3

(Remarks to the Author)

The manuscript is well revised and I have no further question.

Reviewer #4

(Remarks to the Author)

The authors have addressed the suggestions appropriately. Therefore, the manuscript is acceptable for publication.

Reviewer #5

(Remarks to the Author)

I am satisfied with the authors' response. I think that the paper can be accepted now.

Note: Responses of the authors are printed in blue. Changes to the manuscript text (in green) are highlighted by yellow marks. References are numbered according to their appearance in this document and may vary in the main text.

Reviewer #1:

The paper entitled “Purity of Electrodeposited Lithium and its Impact on Lithium Stripping in “Anode-Free” Solid-State Batteries” evaluates the purity of electrodeposited lithium at the interface of steel/Li6PS5Cl solid electrolyte in reservoir-free cell concepts and its influence on lithium stripping in symmetric cells using either Li6PS5Cl or Li6.25Al0.25La3Zr2O12 solid electrolytes. While the research findings are quite interesting and promising, more detailed explanations should be given for a better understanding of the manuscript. The following points need to be addressed prior to publication.

Response: We thank the reviewer for the valuation of our work and the detailed review. We think that our responses will adequately address the open points highlighted by the reviewer, which will further strengthen our work.

Comment/Question 1: In Figure 2a, Li 1s XPS spectra of LiLP, LiHP and LiD show different shapes. Particularly, Li 1s XPS data of LiHP contain 2 overlapping components and another small peak around 55 eV is detectable for LiLP, suggesting the existence of different Li-based species at the surface of LiLP and LiHP foils. This is expected to impact the interfacial compatibility between the lithium foils and solid electrolyte. The authors should deconvolute XPS core peak with the assignment of each component.

Response: We would like to thank the reviewer for the detailed examination of the XPS measurements and the opportunity to strengthen our analysis. On closer inspection, we noticed that the XPS spectra shown were still based on surface measurements of the corresponding lithium foils. Here, the passivation was also removed before the measurement, but only mechanically. For better comparability with the LiD sample, additional measurements were carried out on a cross-section of the respective lithium samples, which were produced using ion polishing, as described in the text. This means that the measurement geometry and the sample preparation now match between all lithium samples, which further confirms the comparability of the samples and thus the resulting conclusions. These spectra have now been included correctly. We would like to note at this point that the quantification shown in the initial version was already based on the spectra from samples prepared by ion polishing. We would like to emphasize that despite the careful preparation method chosen here, lithium shows a degradation

layer on the surface in all cases. This is unavoidable due to the high reactivity of lithium¹ and can vary depending on different background pressures during preparation or transfer. Please note that the XPS spectra were calibrated to the oxygen O 1s signal instead of adventitious carbon due to the previous ion beam polishing. The experimental part was adjusted accordingly:

“Post referencing to the O 1s signal ($E_B = 529.0$ eV) ensured energy calibration.”

Figure A: Updated version of Figure 2 of the main manuscript.

In addition, we corrected the signal intensity of the ⁷Li⁺ signal in the ToF-SIMS depth profiles due to a measurement artifact (ion detection limitation) at the beginning of the measurement. To clarify, we have added a sentence to the ToF-SIMS experimental part.

“All lithium metal samples were first sputter cleaned in a 300-300 μm² area with 1 keV Ar⁺ ions (total dose density 1·10¹⁷ ions cm⁻²) to remove the native oxidation layer formed on the lithium metal during transfer to the SIMS machine. Please note that the ⁷Li⁺ intensity was corrected

using the ${}^6\text{Li}^+$ intensity. Three surface spectra per sample were recorded on the cross-sections of the electrodeposited lithium (prepared by ion milling) in areas of $50 \times 50 \mu\text{m}^2$ with 128×128 pixels.”

Comment/Question 2: The authors should explain better the sputtering process for XPS analysis (paragraph starting line 235). In S2, how can the authors differentiate Li preferential sputtering and degradation? The apparition of a second Li 1s component at lower BE for 20kV needs to be explained.

Response: We thank the reviewer for allowing us to clarify the sputtering process during XPS measurements. As described, all samples were subjected to surface sputtering during XPS measurements. The workflow is as follows: An XPS spectrum was recorded first. The surface was then sputtered using argon gas clusters with an acceleration voltage of 10 kV for 120s and then the next XPS spectrum was recorded. This process was repeated multiple times. For clarity, we modified the experimental section for the XPS measurements:

“Alternating gas cluster ion bombardment was used to clean the surface in between spectra acquisition. The initial spectrum was recorded without sputtering beforehand. After acquisition, the surface was sputtered for 120 s and the second spectrum was acquired. This process was repeated multiple times. The acceleration voltage of the Ar clusters was either 10 kV or 20 kV.”

After the iterations with an acceleration voltage of 10 kV, one single sputter step with an acceleration voltage of 20 kV was conducted.

“After 360 s of sputtering with an acceleration voltage of 10 kV, the intensity approaches saturation. A subsequent increase of the acceleration voltage of the argon clusters to 20 kV significantly increases the Na 2s signal, indicating preferential sputtering or a mass related effect.^{2,3} Furthermore, increasing the accelerating voltage while keeping the argon cluster size constant could lead to an increased removal of inorganic degradation products during argon cluster sputtering, as shown by Shard and Baker:³ Sputtering at an acceleration voltage of 20 kV leads to an overestimation of the sodium content of around 5.4 at% in Li_{LP} .”

The reason for the changed shape of the Li 1s peak in Figure S2 could be a shift in the ratio of organically and inorganically sputtered species. Based on the work of Shard and Baker, the ratio of inorganic to organic sputtered species increases during cluster sputtering with argon clusters at approximately constant cluster size and increasing accelerating voltage (*i.e.* increasing energy per cluster atom).³ Thus, as mentioned by the reviewer, more inorganic degradation products such as LiOH or Li_2O could be removed during the sputtering process with an

accelerating voltage of 20 kV, which increases the proportion of metallic lithium and is thus the cause of the changed peak shape. To clarify, we added a sentence to the XPS discussion under Figure 2 and adjusted the caption of Figure S2:

*“As exemplary shown for the Li_{LP} sample in **Figure S2**, the intensity for the Na 2s signal increases with increasing sputtering time. After 360 s of sputtering with an acceleration voltage of 10 kV, the intensity approaches saturation. A subsequent increase of the acceleration voltage of the argon clusters to 20 kV significantly increases the Na 2s signal, indicating preferential sputtering or a mass related effect.^{2,3} Furthermore, increasing the accelerating voltage while keeping the argon cluster size constant could lead to an increased removal of inorganic degradation products during argon cluster sputtering, as shown by Shard and Baker.³ Sputtering at an acceleration voltage of 20 kV leads to an overestimation of the sodium content of around 5.4 at% in Li_{LP} . The surfaces of the samples were therefore sputtered using an acceleration voltage of 10 kV in between measurements. The following quantification is based on the XPS spectra after sputtering with an acceleration voltage of 10 kV.³”*

*“**Figure S2**: The influence of argon cluster sputtering on the acquired XP spectra. The image on the left displays the acquired XP spectra after the given sputtering times and acceleration voltages. Three sputter steps with an acceleration voltage of the argon clusters of 10 kV were conducted. After that, one additional sputtering step with an acceleration voltage of 20 kV was conducted. The image on the right displays the calculated sodium content (in at%) in dependence of the sputtering time. The change in shape of the acquired spectra and the change in calculated sodium concentration during sputtering with 10 kV indicates that lithium is preferentially sputtered, leading to an apparent increase in sodium. Moreover, Shard and Baker³ have demonstrated that increasing the accelerating voltage while maintaining a constant argon cluster size can enhance the removal of inorganic degradation products during argon cluster sputtering, which could explain the different shape of the XPS spectrum after the final sputtering step with an acceleration voltage of 20 kV.”*

The accompanying increase in the Na 1s signal could thus indicate a layered structure similar to that found during ToF-SIMS depth profiling. Here, a sodium-rich layer was also found under the degradation products LiOH and Li₂O.

To emphasize these potentially complementary results, we have included the following sentence in the discussion of Figure 2:

“Despite ion beam polishing under vacuum and inert sample transfer conditions, a surface passivation layer is visible on all samples, with an increased intensity of Li_2OH^+ and Li_2O^+ . Notably, LiLP shows a sodium-rich layer beneath this oxygen-rich layer. We note that sodium segregation out of lithium metal has early been reported by Schily et al. and Powell et al.^{4,5} This layered structure could also be the reason for the strong increase in the Na 2s signal found following the sputtering step with 20 kV during the XPS analysis, in which more inorganic species such as LiOH and Li_2O are sputtered, exposing a sodium rich layer underneath.³ In contrast, LiHP and LiD show no significant Na^+ intensity, already indicating a reduced sodium content. To avoid an influence of the passivation layer, quantification was performed on all samples after cleaning by sputtering.”

However, as the resulting sodium concentration appears unreasonably high at 5.4 at%, the quantification was carried out based on the spectra after sputtering with an acceleration voltage of 10 kV.

Comment/Question 3: For ToF-SIMS depth-profile, the authors should explain better the depth profiles especially for LiLP : why the sodium signal is increasing after the lithium one is decreasing? Does this mean the sodium is located underneath the lithium? The spatial distribution in depth of the different species should be more explained/developed. Moreover, it would be interesting to investigate the full positive and negative ToF-SIMS mass spectra to see if others species than Li^+ , Na^+ , Li_2OH^+ and Li_2O^+ can be of interest; indeed, the authors focused on Na impurities but what about the other species formed at the interfaces during the different processes.

Response: We thank the reviewer for the question regarding the ToF-SIMS measurements. As also concluded by the reviewer, we assume that the sodium-rich layer is located under a lithium passivation layer. At this point, we would like to refer to the paragraph in lines 276-278, in which we refer to this layered structure. Please note that we have added a sentence regarding the XPS analysis in line with Comment/Question #2.

“Despite ion beam polishing under vacuum and inert sample transfer conditions, a surface passivation layer is visible on all samples, with an increased intensity of Li_2OH^+ and Li_2O^+ . Notably, LiLP shows a sodium-rich layer beneath this oxygen-rich layer. We note that sodium segregation out of lithium metal has early been reported by Schily et al. and Powell et al.^{4,5} This layered structure could also be the reason for the strong increase in the Na 2s signal found following the sputtering step with 20 kV during the XPS analysis, in which more inorganic species such as LiOH and Li_2O are sputtered, exposing a sodium rich layer underneath.³ In

contrast, Li_{HP} and Li_D show no significant Na⁺ intensity, already indicating a reduced sodium content. To avoid an influence of the passivation layer, quantification was performed on all samples after cleaning by sputtering. “

We would also like to thank the reviewer for the question regarding other types of contamination. We agree with the reviewer that other impurities should have a similar effect as sodium impurities, e.g. reduce the stripping capacity of lithium electrodes. As described by the reviewer, we focus on the main impurity sodium in the manuscript.⁶ However, we have also investigated other alkali and alkaline earth metal impurities such as K, Mg and Ca. Figure C shows the depth profile from Figure 2 in the manuscript with the signals for K⁺, Mg⁺ and Ca⁺ secondary ions.

Figure C: ToF-SIMS depth profile of the Li_{LP} sample as in Figure 2 of the manuscript. Additionally, the signals for K⁺, Mg⁺ and Ca⁺ are shown. The lower part of the figure is an enlargement of the upper part, as indicated by the red dashed box.

The depth profile shows that the K⁺, Mg⁺ and Ca⁺ signal intensity is approximately 3 orders of magnitude lower than that of Na⁺. For this reason, we have not included them in the discussion within the manuscript. Interestingly, these signals also show an increase in intensity below the

lithium degradation layer. We would like to add that, if the K^+ signal of the lithium foil is compared with the K^+ signal on the LPSCl surface after stripping (please see Figure 3 in the manuscript), the formation of an impurity layer is once again underpinned.

Comment/Question 4: It is recommended to provide the XRD spectrum of the as-synthesized LLZO phase as well as a brief discussion regarding the phase purity.

Response: We thank the reviewer for highlighting the importance of using phase-pure LLZO. As suggested, the results of a powder XRD measurement of an as-synthesized LLZO sample can be seen in Figure D. No significant impurity fractions could be found. However, since the focus of the work is not on the solid electrolytes, we refrain from showing the diffractogram shown here in the SI.

Figure D: Powder diffractogram of the as-prepared LLZO solid electrolyte. The ticks for the cubic phase are given underneath.

Comment/Question 5: The Li^+ ionic conductivity of the prepared LLZO phase should also be provided.

Response: According to the suggestion of the reviewer, the lithium conductivity of the LLZO pellet used in the manuscript was calculated based on the impedance measurement before the stripping experiment with two lithium electrodes. This is possible because the electrodes were applied according to the approach of Krauskopf et al.⁷ Thus, the contribution at the $Li|LLZO$

interfaces is negligibly small, and both the bulk and the grain boundary contribution can be extracted from the impedance measurement. The corresponding impedance spectrum is shown in Figure E. The conductivity calculated from this is 0.81 mS cm^{-1} and is therefore in agreement with literature values.⁷ However, as the focus of the work is not on the solid electrolytes, we also refrain from including this conductivity calculation in the SI. Please note that the origin of the low-frequency part of the impedance spectrum ($< 100 \text{ Hz}$) has not yet been fully clarified. In literature, this is often associated with diffusion processes within the electrodes and is therefore neglected for conductivity calculation.⁷ Interestingly, this low-frequency contribution occurs independently of the lithium sample used (compare Figure S11).

Figure E: Impedance spectrum of an as-prepared LiLP|LLZO|LiLP symmetrical cell. The two contributions associated with bulk and grain boundary (GB) transport are highlighted with semicircles. The calculated conductivity is given in the table on the right.

Comment/Question 6: Please add S to XPS in lines 220, 223, 228, 236, 245, 256, 326, 331, 335

Response: As suggested by the reviewer, we replaced "XP spectra" with "XPS spectra" in all appropriate places.

Comment/Question 7: The authors should organize better the different parts of the article by adding subtitles (ex: for EDX...)

Response: At this point we disagree with the reviewer. According to the standards of Nature Communications, the manuscript is formatted accordingly and organized into several subtopics instead of method subsections. However, to improve the clarity of the manuscript, we rearranged the paragraph on page 11.

"The two lithium samples, LiLP and LiHP were additionally analyzed using EDX to obtain information about the distribution of sodium within the lithium foil. The lithium samples were

attached to an LPSCl pellet, a capacity of 5 mAh cm^{-2} was stripped from one of the two lithium electrodes, and a crater was milled in the stripping electrode by using a focused ion beam. The EDX analysis reveals sodium precipitates within the lithium matrix of Li_{LP} (see **Figure S4**). In contrast, no such precipitates and almost no signal for sodium in general were found for Li_{HP} (see **Figure S5**). These precipitates likely stem from lithium metal production from the melt, in which sodium impurities are inevitably present. Based on the binary phase diagram, lithium and sodium form a monotectic at low sodium content (see **Figure S6**).⁸ As lithium shows virtually no or minor solubility for sodium, the formation of sodium precipitates is to be expected.”

This makes it clear in the very first sentence that the results of EDX measurements will be discussed below and we hope that the reader will be able to follow along more easily. The paragraph on page 13 regarding EDX measurements has also been revised to improve the differentiation between the various measurement methods, and it now reads like this:

“To validate the accumulation of impurities at the solid electrolyte interface, additional experiments at an $\text{Li}_{\text{LP}}|\text{Li}_{6.25}\text{Al}_{0.25}\text{La}_3\text{Zr}_2\text{O}_{12}$ (LLZO) interface were conducted. LLZO is studied as a potential separator SE for SSBs. It is practically stable in contact with lithium metal, allowing us to rule out degradation at the $\text{Li}|\text{LLZO}$ interface in this case.⁹ One electrode of a symmetrical $\text{Li}_{\text{LP}}|\text{LLZO}|\text{Li}_{\text{LP}}$ cell was electrochemically stripped until pores formed. This allowed the stripped Li_{LP} electrode to be peeled off and the surface of the LLZO pellet to be analyzed.⁷ Here, too, an enrichment of sodium on the surface of LLZO could be detected, which is even visible to the naked eye as a dark film on the SE. The results of this investigation using ToF-SIMS and XPS are summarized in **Figure S7**. In addition, EDX measurements were carried out at the edge of the electrode surface, which also show sodium accumulation for Li_{LP} in the area of the previous electrode contact. The results of the EDX analysis can be seen in **Figure S8**.”

Comment/Question 8: The quality of the image in Figure S6 should be improved

Response: We thank the reviewer for pointing out the low quality of Figure S6. We have replaced the image with a higher resolution version, which is shown in Figure F.

Figure F: Updated version of Figure S6 in the supplementary information.

“Figure S6: Phase diagram of the binary Li-Na system. The magnified section on the right highlights the low sodium regime, revealing a monotectic at a sodium content of 3.4 at%. As can be seen from the phase diagram, only a very low amount of sodium can be dissolved homogeneously in solid lithium at room temperature. Accordingly, lithium production from a melt, inevitably containing sodium as an impurity, leads to sodium precipitation. The amount, size and distribution of the sodium precipitates will depend on the specific sodium content and processing route. Reproduced and modified from Zhang et al.⁸. Copyright 2003 Elsevier.”

Comment/Question 9: The monotectic phase could be more explained/developed (it is not clear on the S6 figure)

Response: We thank the reviewer for the opportunity to elaborate on the monotectic phase. In general, a monotectic transformation takes place when a liquid phase decomposes into both a solid phase and a new liquid phase.¹⁰ In the case of lithium and sodium, the monotectic isotherm is at a temperature of around 445 K. The monotectic point is positioned at a mole fraction of sodium of approximately 0.03.⁸ This section of the phase diagram is therefore enlarged on the right in Figure E.

Comment/Question 10: Figure S8: the authors should mention that the intensity signal is not comparable between the two images otherwise the proportion of Na seems to be higher for LiHP

Response: We thank the reviewer for pointing out that the sodium intensities in Figure S8 are not comparable, and we agree with the reviewer. To avoid misunderstandings, we have edited the corresponding figure caption, and it now reads as follows:

“Figure S8: Top-view SEM and EDX analysis of a LLZO surface after being in contact with either Li_{LP} (top images) / Li_{HP} (bottom images) during lithium stripping (see Figure 4). The area, where the lithium electrode was in contact with the LLZO pellet can be seen in the left SEM images. The EDX map of the Na K_{α1,2} signal on the right shows an increased intensity in the area where the Li_{LP} electrode was in contact with the LLZO pellet. Conversely, for the Li_{HP} electrode a homogeneous signal is visible, being only due to noise. Please note that the intensities of the two Na K_{α1,2} signals have not been normalized with each other and therefore the intensities between the two images are not comparable.”

Comment/Question 11: Line 388: it should be written figure S11 and not S10.

Response: We thank the reviewer for pointing out the incorrect numbering. The reference has been adjusted accordingly and now points to the correct image **Figure S11**.

“A similar conclusion can be drawn for Li|LLZO interfaces, which are prepared at an isostatic pressure of 400 MPa, ensuring optimal interfaces with minimal resistance in both cases (compare Figure S11).”

Comment/Question 12: The paragraph from lines 427 to 435 is not clear and needs to be revised.

Response: We thank the reviewer for allowing us to clarify our statements in the specific paragraph. To analyze the stripping capacity of the electrochemically deposited lithium, a lithium layer first had to be electrochemically deposited. As described, 5 mAh cm⁻² of lithium was deposited on a steel|LPSCl interface and then dissolved again.

“To investigate the stripping performance of Li_D, a lithium layer with a capacity of 5 mAh cm⁻² was initially electrodeposited at a steel|LPSCl interface with a current density of 100 μA cm⁻² at an external pressure of 15 MPa. The voltage profile during electrodeposition is displayed in Figure S9. Subsequently, the external pressure was released, the current direction was reversed, and lithium was stripped until a pore formation occurred.”

As shown in Figure 4, the stripping capacity of Li_D amounts to 3.74 mAh cm⁻² (equals roughly 75% of the initially electrodeposited 5 mAh cm⁻²) and exceeds that of the other two samples Li_{LP} and Li_{HP}. On the one hand, Li_D has the highest purity, but we would like to point out that

purity is not the only parameter that differs between Li_{LP} and Li_{D} . In particular, a higher contact area than the geometrically assumed one could lead to more lithium being stripped. To clarify, we adjusted the paragraph in the following way:

*“In the case of Li_{D} , an even higher stripping capacity of 3.74 mAh cm^{-2} is achieved, corresponding to 75% of the previously electrodeposited 5 mAh cm^{-2} of lithium. However, this increase in stripping capacity cannot be exclusively attributed to a higher purity of Li_{D} , as other parameters also change. On the one hand, the $\text{Li}_{\text{D}}|\text{LPSCl}$ interface contact could be more homogeneous due to the electrochemical deposition than is the case for the conventionally prepared $\text{Li}|\text{LPSCl}$ interfaces. In addition, the actual contact area between Li_{D} and LPSCl could be larger than the geometrically assumed one, since deposition within the SE could also have occurred during initial electrodeposition. This hypothesis is supported by a cross-section prepared on the $\text{Li}_{\text{D}}|\text{LPSCl}$ interface of a similar sample, revealing dendritic structures, i.e. lithium that has been deposited inside the solid electrolyte (see **Figure S14**).”*

Comment/Question 13: Line 532: it should be written Figure 6b and not 6c.

Response: We would like to thank the reviewer once again for the detailed review. The reference has been changed accordingly and now correctly points to **Figure 6b**.

*“The aforementioned impurities in case **Figure 6b** lead to an increase of the local current density, as they block the local ion flux, decreasing the active electrode area.”*

Reviewer #2:

Response: We thank the reviewer for the participation in the co-review program and for the contribution to the corresponding review.

Reviewer #3:

In this paper, purity of electrodeposited lithium and its impact on lithium stripping in “anode-free” solid-state batteries are discussed. Although the purity of lithium metal anodes is usually ignored in lithium metal batteries, the impact of impurities on electrochemical performance in anode-free systems still requires deeper consideration. Here are some questions that need to be clarified and answered by the authors.

Response: We would like to thank the reviewer for the thorough review and for highlighting that the role of impurities in lithium metal batteries has not been sufficiently considered to date. We believe that the open points can be adequately answered, further improving the manuscript.

Comment/Question 1: The influence of cathode impurities on lithium stripping is mentioned in the abstract (it shows how impurities at the positive electrode affect stripping capacity.....) and conclusion (We also like to note that true reservoir-free SSB cells will most probably be unaffected by any impurity accumulation at the solid electrolyte interface.....). However, the manuscript does not provide any reference to the test results regarding the impact of cathode impurities on lithium deposition and stripping. In addition, the filtration capacity of solid-state electrolytes is also mentioned by the authors. Therefore, whether the impurities in the cathode will have an effect on the lithium metal anode requires more careful clarification by the authors.

Response: We thank the reviewer for addressing the ambiguous wording. As we understand it, the reviewer is referring to the following sentence in lines 70-72:

“Research on cathode active materials shows that lithium impurities can significantly affect their electrochemistry.¹¹⁻¹³ Yet, little is known about the impact of lithium metal purity on its electrochemical performance in SSBs.”

To be clear, what we are referring to here is that impurities in common cathode active materials (CAMs) such as NCM, LNO and LCO have an impact on the electrochemical properties of these CAMs. This in turn suggests that the same applies to the lithium metal electrode. We do not connect the impurity content in the cathode with the content in the anode. To clear up the misunderstanding, we have adjusted the wording in this paragraph as follows in the hope that it will be more understandable. In fact, one could argue that impurities in the cathode electrolyte might transfer to the anode once the separator electrolyte is not 100% selective for Li ions. While this is an interesting question, it is beyond the scope of the present manuscript.

“Research on cathode active materials reveals that contaminants within these materials can significantly influence their electrochemical properties.¹¹⁻¹³ Yet, little is known about the impact of lithium metal purity on its electrochemical performance in SSBs.”

Comment/Question 2: The key scientific issues of the manuscript require the authors to consider. Through the experiments, the authors conclude that LiD has the highest purity. In the anode-free system selected by the authors, all lithium metal originates from LiD, which is virtually free of impurities. Therefore, the necessity of the topic of “purity of electrodeposited

lithium and its impact on lithium stripping in anode-free solid-state batteries” needs further explanation.

Response: We would like to thank the reviewer for the comment. While we consider a major contribution of this work to be the initial demonstration of the purity of electrodeposited lithium in “anode-free” cells, we agree with the reviewer that the subsequent analysis focuses on lithium metal anodes and the impact of their purity in general. To make this clear directly in the title of the manuscript, we adjusted the title as follows:

“Purity of Lithium Metal Anodes and its Impact on Lithium Stripping in Solid-State Batteries”

Comment/Question 3: Although the authors also state that minimal lithium reservoir is used to compensate for irreversible lithium losses, this impurity preferentially affects the lithium deposition. Therefore, the influence of impurities on lithium deposition needs to be discussed by the authors. And how does LiD affect the stripping behavior after it is deposited in lithium containing impurities?

Response: We would like to thank the reviewer for once again emphasizing the importance of considering impurities. We agree with the statement that impurities in existing lithium reservoirs also have an impact on the deposition of lithium on these reservoirs. We like to clarify this highlighting the corresponding paragraph.

“In practice, SSBs are often investigated with a minimal lithium reservoir on the negative electrode, aiming for the compensation of irreversible lithium loss. While these cells are still promising to achieve higher energy densities,¹⁴ the role of the impurity content in the lithium reservoir is not known.”

One of the key points of this statement is to emphasize that lithium purity is an important parameter that - as mentioned at the beginning - has so far been overlooked. However, within this study, we focus on lithium dissolution. Nevertheless, we agree with the reviewer that subsequent work should carry out more in-depth analyses of the influence of impurities on lithium deposition. The same applies to the “targeted contamination” of deposited lithium. Initial work has shown that alloy-forming materials such as gold or silver can have a positive effect on deposition.^{15,16} Nevertheless, a deeper understanding of this, also in subsequent dissolution cycles, should be sought. To trigger more research in this direction, we have included a new paper by Yoon et al.¹⁷ in the discussion at the appropriate point, which outlines the influence of sodium impurities on the deposition process:

“This behavior aligns with recent findings by Mann et al., who reported the formation of a 3D sodium scaffold during the stripping process from a monotectic lithium-sodium mixture with 3 at% Na prepared through cold work.¹⁸ In contrast to our findings, the authors report improved lithium stripping kinetics which they attribute to the 3D structured Na interlayer formed during the first stripping step. A direct comparison with these results is difficult, as only garnet SE was used and as most experiments were run at 60 °C. Similarly, Yoon et al. also investigated the influence of several at% of sodium in lithium during stripping. Under pressure, an increase in stripping capacity was observed with increasing sodium content, which Yoon et al. attributed to the greater plastic deformability of the sodium-lithium mixture.¹⁹ During stripping, there was also an accumulation of sodium at the solid electrolyte interface observed. As in the case of Mann et al., the resulting sodium layer appears to avoid current focusing in the subsequent plating step, as it keeps contact between the electrode and the solid electrolyte. Additionally, Park et al. reported an improved lithium stripping by introducing a Na-K interlayer at the Li|LLZTO interface.²⁰ More work will be necessary to resolve this apparent contradiction. We suggest that small Na atom fractions, typical for a homogeneously dissolved impurity, will deteriorate the interface conformity, while a larger fraction of sodium, dispersed as a second phase, may form a thick interlayer.”

Comment/Question 4: The selected ranges of 1 and 2 in Figure S5 are not consistent with the shape of the local magnification figures, and the authors need to accurately select the magnification range.

Response: We would like to thank the reviewer for the attentive reading. The section that was incorrectly specified as too small has been corrected and the entire EDX measurement recorded is now displayed correctly, as displayed in Figure G.

Figure G: Updated version of Figure S5 in the supplementary information.

Comment/Question 5: In P10 the authors speculate that the impurity may originate from the diffusion of Na from the solid-state electrolyte to the electrodeposited lithium. Instead in P11 the authors emphasize “we conclude that only lithium ions can migrate through LPSCI”. These are contradictory results.

Response: We would like to thank the reviewer for pointing out this potentially misleading statement. We like to clarify that our statement is based on the (theoretical) possibility that sodium impurities contained in the solid electrolyte (e.g. from contaminated precursors during synthesis) chemically interact with the deposited lithium layer. However, it is not the case that these impurities can migrate through the solid electrolyte during electrochemical deposition.

Comment/Question 6: In the XPS analysis of Figure 3a, the authors analyzed the occurrence of sodium accumulation on the surface of the solid-state electrolyte. However, the peak ratio of

Na 2s to Li 1s does not seem to differ significantly from that in LiLP (Figure 2a). How do the authors account for sodium enrichment? This needs to be clarified by the authors. Similarly, ToF-SIMS results (Figure 3b) are virtually indistinguishable from pure LiLP (Figure 2c). Therefore, the sodium-rich layer and sodium accumulation require more convincing evidences.

Response: We thank the reviewer for raising this point. As discussed in the answer to Comment/Question 1 of Reviewer #1, we adjusted the respective XPS spectra. We did this because the spectra shown showed surface measurements of the respective lithium foils. Instead, the spectra now shown were recorded on cross-sections that were produced in the same way as the cross-section of the LiD sample, ensuring better comparability. The spectra recorded on cross-sections show a lower ratio between the Na 2s and Li 1s peaks, further strengthening the hypothesis of a sodium accumulation on the LPSCl surface during the course of the work (compare Figure 3). Differences from surface measurements probably come from the different measurement geometries and the different sample preparation. With regard to the ToF-SIMS measurements, we like to point out that the fluence in the measurement of the LPSCl surface is one order of magnitude higher compared to the measurement of the lithium sample LiLP. Accordingly, it can be concluded that the sodium-rich layer on the LPSCl surface is significantly thicker than the sodium layer on the lithium foil itself. Please note that, as discussed in the response to Comment/Question 1 of Reviewer #1, the $^7\text{Li}^+$ intensity has been corrected due to a measurement artefact. Furthermore, we would like to refer the reviewer to the response to Comment/Question 3 of Reviewer #1: In addition to the Na^+ signal, a significant increase in the K^+ signal can be observed on the LPSCl surface, which also confirms the accumulation of impurities at the interface to the solid electrolyte. This result is confirmed by additional measurements on an LLZO surface after contact with LiLP during stripping (see Figure S7). In addition to the ToF-SIMS and XPS measurements, the EDX measurements following a stripping experiment with subsequent removal of the lithium electrode from Figure S8 also confirm an accumulation of sodium at the interface to the solid electrolyte.

Comment/Question 7: Why does the intensity of LiHP noise in Figure S8 even exceed that of Na in LiLP?

Response: We would like to thank reviewer for pointing out this misunderstanding. As highlighted by Reviewer 1 in Question/Comment 8, the two intensities cannot be compared between the two images. To make this clear, we have adjusted the caption accordingly.

“Figure S8: Top-view SEM and EDX analysis of a LLZO surface after being in contact with either LiLP (top images) / LiHP (bottom images) during lithium stripping (see Figure 4). The

area, where the lithium electrode was in contact with the LLZO pellet can be seen in the left SEM images. The EDX map of the Na $K_{\alpha 1,2}$ signal on the right shows an increased intensity in the area where the Li_{LP} electrode was in contact with the LLZO pellet. Conversely, for the Li_{HP} electrode a homogeneous signal is visible, being only due to noise. **Please note that the intensities of the two Na $K_{\alpha 1,2}$ signals have not been normalized with each other and therefore the intensities between the two images are not comparable.**”

Comment/Question 8: In Figure 4, the authors are advised to repeat the experiment several times to provide statistical results, thereby ensuring the accuracy of the experimental results.

Response: We would like to thank the reviewer for emphasizing the importance of reproducibility and agree that statistics are necessary to substantiate experimental data. For this reason, several cells were analyzed for the respective symmetrical cells as described in the experimental.

“To ensure reproducibility, multiple cells were analyzed.”

To be precise, triplicates of the different symmetrical cells were built and analyzed. The stripping curves for the different cell systems, i.e. $Li_{LP}|LPSCl|Li_{LP}$, $Li_{HP}|LPSCl|Li_{HP}$ and $Li_{LP}|LPSCl|Li_D$, can be seen in Figure H.

Figure H: Stripping curves of the cell systems examined using LPSCl as solid electrolyte. Three cells of type $\text{Li}_{\text{LP}}|\text{LPSCl}|\text{Li}_{\text{LP}}$, $\text{Li}_{\text{HP}}|\text{LPSCl}|\text{Li}_{\text{HP}}$ and $\text{Li}_{\text{LP}}|\text{LPSCl}|\text{Li}_{\text{D}}$ are shown. The respective cell system is shown schematically next to the corresponding stripping curves.

Despite minor deviations within the respective cell systems, it is clear that the stripping capacities of $\text{Li}_{\text{D}} > \text{Li}_{\text{HP}} > \text{Li}_{\text{LP}}$. The drawn conclusions of the manuscript are therefore supported. Interestingly, the stripping capacity of approximately 1 mAh cm^{-2} for Li_{HP} is in a similar range to the stripping capacities achieved for lithium from previous work.^{21–23} This is consistent with the observations made here that the electrode is the limiting factor during stripping.

Although there are minor deviations in the stripping capacity within the different cell systems, the trend between the different lithium samples is evident in every case and is confirmed by the statistics. At this point we like to mention that the stripping curve marked "Electrodeposited at $50 \mu\text{A cm}^{-2}$ " was deposited at a current density of $50 \mu\text{A cm}^{-2}$. All layers were dissolved at a current density of $100 \mu\text{A cm}^{-2}$. This serves for ongoing investigations into the influence of lithium deposition on subsequent lithium dissolution. Nevertheless, it is clear that Li_{D} provides a significantly higher stripping capacity than the other lithium samples.

The stripping curves for the LLZO cells, i.e. $\text{Li}_{\text{LP}}|\text{LLZO}|\text{Li}_{\text{LP}}$ and $\text{Li}_{\text{HP}}|\text{LLZO}|\text{Li}_{\text{HP}}$ are shown in Figure I.

Figure I: Stripping curves of the cell systems examined using LLZO as solid electrolyte. Three cells of type Li_{LIP}|LLZO|Li_{LIP} and Li_{HP}|LLZO|Li_{HP} are shown. The respective cell system is shown schematically next to the corresponding stripping curves.

Also using LLZO, the trend that the stripping capacity of Li_{HP} is increased compared to Li_{LIP} is confirmed.

For clarity, we have adapted the relevant text in the experimental part as follows:

“To ensure reproducibility of the observed trends, a set of three cells was examined for each symmetrical cell system.”

Reviewer #4:

This study studied in detail the effect of commercially available lithium metal foils and "Anode-Free" lithium electrodeposition on the stripping capacity of the anode. If it can be put into commercial use, it will make it possible to create a rechargeable battery with a higher energy density than existing LIBs. To this end, it shows a new path from the perspective of the effect of impurities in the electrode, in terms of how to achieve reversible and stable charging and discharging of the Li metal anode. However, in the final discussion, it seems that a lot of information has been omitted in order to conceptualize the topic, and many questions remain.

Response: We thank the reviewer for the positive evaluation of our work and agree that lithium purity is an important parameter to achieve reversible and stable cycling of lithium metal electrodes.

Comment/Question 1: If possible, could a more detailed discussion be considered from both sides of the solid electrolytes $\text{Li}_6\text{PS}_5\text{Cl}$ (LPSCl) and $\text{Li}_{6.25}\text{Al}_{0.25}\text{La}_3\text{Zr}_2\text{O}_{12}$ (LLZO) that were discussed in this study? This is because, for example, the SEM analysis of the stripped Li electrode in Fig. S12 does not include a discussion of the differences between the solid electrolytes in Fig. 6. I feel that the impact is significant, so I would like you to consider this.

Response: We agree with the reviewer that, in addition to the lithium metal electrode, the choice of solid electrolyte also has an influence on cell performance. We would like to clarify that the electrodes examined in Figure S12 were both investigated in combination with LLZO as solid electrolyte to ensure comparability of the electrode morphologies after stripping. We would like to add that the choice of solid electrolyte is less important for electrode characterization than the electrode itself, as shown by the results of this work and previous work.²⁴

Comment/Question 2: In addition, the short circuit when using a lithium anode with LPSCl as the solid electrolyte is described in Fig. 4. This suggests the possibility that Li is penetrating the solid electrolyte. Do these correspond to the work of Porz (10.1002/aenm.201701003) or the different claims of Shen (10.1021/acseenergylett.8b00249) or Han (10.1038/s41560-018-0312-z), etc., which of the different claims do you correspond to? Is it difficult to consider it changes depending on the rigidity of the solid electrolyte? When these considerations are added, I feel that the discussion in Fig. 6 also expands.

Response: We would like to thank the reviewer again for picking up on the solid electrolyte influence. We agree that the mechanical properties, in particular, differ significantly between the sulfide $\text{Li}_6\text{PS}_5\text{Cl}$ solid electrolyte and the oxide solid electrolyte $\text{Li}_{6.25}\text{Al}_{0.25}\text{La}_3\text{Zr}_2\text{O}_{12}$, potentially influencing the process of dendrite formation. However, we like to point out that the two materials also differ in terms of their electrochemical stability towards lithium and their conductivity - both electronically and ionically. It is therefore difficult to draw reliable conclusions on the dendrite formation mechanisms mentioned without an in-depth analysis of the two solid electrolyte materials. Since this study focuses on the characterization of the lithium metal anode, the detailed consideration of the various solid electrolyte materials is, in our opinion, out of scope.

In general, my comments do not deny the main point of the paper, but are a request for a little more in-depth analysis. I really like the approach of this study.

Response: We would like to thank the reviewer once again for appreciating our work.

Reviewer #5:

This paper provides new insight into how to improve performance of all-solid-state batteries with lithium anode. The paper developed a new mechanism that impurity will accumulate at the lithium/solid electrolyte interface and deteriorate performance. The analysis is comprehensive. I have a couple of minor questions.

Response: We would like to thank the reviewer for appreciating our work and for the comments provided.

Comment/Question 1: Could authors repeat the stripping experiment to show that the difference in stripping capacity is statistically significant?

Response: We would like to thank the reviewer for also taking up the importance of reproducibility. This was also highlighted by Reviewer #3 in Question/Comment 8. We therefore like to refer the reviewer to the above given answer. A set of 3 cells was used as the basis for the conclusions drawn. The corresponding stripping curves can be seen in Figure H and Figure I.

Comment/Question 2: Alloying lithium with other metal is often considered an effective strategy to improve the reversibility of lithium metal anode in all-solid-state batteries. So why is impurity not good, which can be considered as an extreme case of alloying? Is it because Na and Li do not alloy? I suggest authors discuss this point. For example, if Zn or In is intentionally doped inside as impurity, will stripping capacity increase or not? This could help better understand this phenomenon.

Response: We would like to thank the reviewer for raising the question regarding composite or alloy anodes. In fact, targeted contamination of lithium anodes can lead to improved performance. For example, the incorporation of magnesium into lithium can contribute to increased stripping capacity, as shown by Krauskopf et al.²³ Lithium deposition can also be homogenized by alloy-forming materials such as gold, as shown by Haslam et al.¹⁵ Even sodium impurities appear to have a positive influence on lithium metal electrodes with a suitable composition and measurement parameters.^{17,18} As suggested by Yoon et al., differences in

mechanics, for example, could also lead to improved electrode performance. To address this, we have added a sentence at the end of our discussion:

“As suggested by Yoon et al., differences in mechanical properties could also be the cause of the increased stripping capacities of lithium-sodium electrodes.¹⁷”

Whether Zn or In function in a similar way is an interesting question that should be investigated in subsequent work. However, the anode performance also depends on pressure, temperature, and other external measurement parameters making it difficult to give a definitive answer.

Literature

- (1) Otto, S. K.; Moryson, Y.; Krauskopf, T.; Pepler, K.; Sann, J.; Janek, J.; Henss, A. In-Depth Characterization of Lithium-Metal Surfaces with XPS and ToF-SIMS: Toward Better Understanding of the Passivation Layer. *Chemistry of Materials* **2021**, *33* (3), 859–867. <https://doi.org/10.1021/acs.chemmater.0c03518>.
- (2) Sano, N.; Bellew, A.; Blenkinsopp, P. Comparing Sputter Rates, Depth Resolution, and Ion Yields for Different Gas Cluster Ion Beams (GCIB): A Practical Guide to Choosing the Best GCIB for Every Application. *Journal of Vacuum Science & Technology A* **2023**, *41* (5). <https://doi.org/10.1116/6.0002864>.
- (3) Shard, A. G.; Baker, M. A. Practical Guides for X-Ray Photoelectron Spectroscopy: Use of Argon Ion Beams for Sputter Depth Profiling and Cleaning. *Journal of Vacuum Science & Technology A* **2024**, *42* (5). <https://doi.org/10.1116/6.0003681>.
- (4) Powell, G. L.; Clausing, R. E.; McGuire, G. E. Sodium Segregation onto a Lithium Metal Surface. *Surf Sci* **1975**, *49* (1), 310–314. [https://doi.org/10.1016/0039-6028\(75\)90345-3](https://doi.org/10.1016/0039-6028(75)90345-3).
- (5) Mundy, J. N.; McFall, W. D. Comparison of the Isotope Effect for Diffusion of Sodium and Silver in Lithium. *Phys Rev B* **1973**, *7* (10), 4363–4370. <https://doi.org/10.1103/PhysRevB.7.4363>.
- (6) Addison, C. C. *The Chemistry of the Liquid Alkali Metals*; 1984.
- (7) Krauskopf, T.; Hartmann, H.; Zeier, W. G.; Janek, J. Toward a Fundamental Understanding of the Lithium Metal Anode in Solid-State Batteries - An Electrochemo-Mechanical Study on the Garnet-Type Solid Electrolyte Li_{6.25}Al_{0.25}La₃Zr₂O₁₂. *ACS Appl Mater Interfaces* **2019**, *11* (15), 14463–14477. <https://doi.org/10.1021/acsami.9b02537>.
- (8) Zhang, S.; Shin, D.; Liu, Z. K. Thermodynamic Modeling of the Ca-Li-Na System. *CALPHAD* **2003**, *27* (2), 235–241. <https://doi.org/10.1016/j.calphad.2003.09.001>.
- (9) Connell, J. G.; Fuchs, T.; Hartmann, H.; Krauskopf, T.; Zhu, Y.; Sann, J.; Garcia-Mendez, R.; Sakamoto, J.; Tepavcevic, S.; Janek, J. Kinetic versus Thermodynamic Stability of LLZO in Contact with Lithium Metal. *Chemistry of Materials* **2020**, *32* (23), 10207–10215. <https://doi.org/10.1021/acs.chemmater.0c03869>.

- (10) Hu, G.; Cai, X.; Rong, Y. *Phase Transformation and Properties*; De Gruyter, 2020. <https://doi.org/10.1515/9783110495379>.
- (11) Binder, J. O.; Culver, S. P.; Pinedo, R.; Weber, D. A.; Friedrich, M. S.; Gries, K. I.; Volz, K.; Zeier, W. G.; Janek, J. Investigation of Fluorine and Nitrogen as Anionic Dopants in Nickel-Rich Cathode Materials for Lithium-Ion Batteries. *ACS Appl Mater Interfaces* **2018**, *10* (51), 44452–44462. <https://doi.org/10.1021/acsami.8b16049>.
- (12) Reimers, J. N.; Dahn, J. R.; von Sacken, U. Effects of Impurities on the Electrochemical Properties of LiCoO₂. *J Electrochem Soc* **1993**, *140* (10), 2752–2754. <https://doi.org/10.1149/1.2220905>.
- (13) Bianchini, M.; Roca-Ayats, M.; Hartmann, P.; Brezesinski, T.; Janek, J. There and Back Again—The Journey of LiNiO₂ as a Cathode Active Material. *Angewandte Chemie - International Edition* **2019**, *58* (31), 10434–10458. <https://doi.org/10.1002/anie.201812472>.
- (14) Burton, M.; Narayanan, S.; Jagger, B.; Olbrich, L. F.; Dhir, S.; Shibata, M.; Lain, M. J.; Astbury, R.; Butcher, N.; Copley, M.; Kotaka, T.; Aihara, Y.; Pasta, M. Techno-Economic Assessment of Thin Lithium Metal Anodes for Solid-State Batteries. *Nat Energy* **2024**. <https://doi.org/10.1038/s41560-024-01676-7>.
- (15) Haslam, C.; Sakamoto, J. Stable Lithium Plating in “Lithium Metal-Free” Solid-State Batteries Enabled by Seeded Lithium Nucleation. *J Electrochem Soc* **2023**, *170* (4), 040524. <https://doi.org/10.1149/1945-7111/accab4>.
- (16) Sandoval, S. E.; Lewis, J. A.; Vishnugopi, B. S.; Nelson, D. L.; Schneider, M. M.; Cortes, F. J. Q.; Matthews, C. M.; Watt, J.; Tian, M.; Shevchenko, P.; Mukherjee, P. P.; McDowell, M. T. Structural and Electrochemical Evolution of Alloy Interfacial Layers in Anode-Free Solid-State Batteries. *Joule* **2023**, *7* (9), 2054–2073. <https://doi.org/10.1016/j.joule.2023.07.022>.
- (17) Yoon, S. G.; Vishnugopi, B.; Nelson, D.; Xiao Bin Yong, A.; Wang, Y.; Sandoval, S.; Thomas, T.; Cavallaro, K.; Shevchenko, P.; Alsaç, E. P.; Wang, C.; Singla, A.; Greer, J.; Ertekin, E.; Mukherjee, P.; McDowell, M. Interface Morphogenesis with a Deformable Secondary Phase in Solid-State Lithium Batteries. January 10, 2025. <https://doi.org/10.26434/chemrxiv-2025-xdzgq>.
- (18) Mann, M.; Schwab, C.; dos Santos, L. C. P.; Spatschek, R.; Fattakhova-Rohlfing, D.; Finsterbusch, M. Improving the Rate Performance of Lithium Metal Anodes: In-Situ Formation of 3D Interface Structures by Mechanical Mixing with Sodium Metal. *Energy Storage Mater* **2025**, *74*, 103975. <https://doi.org/10.1016/j.ensm.2024.103975>.
- (19) Yoon, S. G.; Vishnugopi, B.; Nelson, D.; Xiao Bin Yong, A.; Wang, Y.; Sandoval, S.; Thomas, T.; Cavallaro, K.; Shevchenko, P.; Alsaç, E. P.; Wang, C.; Singla, A.; Greer, J.; Ertekin, E.; Mukherjee, P.; McDowell, M. Interface Morphogenesis with a Deformable Secondary Phase in Solid-State Lithium Batteries. January 10, 2025. <https://doi.org/10.26434/chemrxiv-2025-xdzgq>.
- (20) Park, R. J. Y.; Fincher, C. D.; Badel, A. F.; Carter, W. C.; Chiang, Y. M. Ultrahigh Areal Capacity Li Electrodeposition at Metal-Solid Electrolyte Interfaces under Minimal

- Stack Pressures Enabled by Interfacial Na-K Liquids. *ACS Appl Mater Interfaces* **2023**, *15* (30), 36117–36123. <https://doi.org/10.1021/acsami.3c04297>.
- (21) Singh, D. K.; Henss, A.; Mogwitz, B.; Gautam, A.; Horn, J.; Krauskopf, T.; Burkhardt, S.; Sann, J.; Richter, F. H.; Janek, J. Li₆PS₅Cl Microstructure and Influence on Dendrite Growth in Solid-State Batteries with Lithium Metal Anode. *Cell Rep Phys Sci* **2022**, *3* (9). <https://doi.org/10.1016/j.xcrp.2022.101043>.
- (22) Fuchs, T.; Haslam, C. G.; Moy, A. C.; Lerch, C.; Krauskopf, T.; Sakamoto, J.; Richter, F. H.; Janek, J. Increasing the Pressure-Free Stripping Capacity of the Lithium Metal Anode in Solid-State-Batteries by Carbon Nanotubes. *Adv Energy Mater* **2022**, *12* (26). <https://doi.org/10.1002/aenm.202201125>.
- (23) Krauskopf, T.; Mogwitz, B.; Rosenbach, C.; Zeier, W. G.; Janek, J. Diffusion Limitation of Lithium Metal and Li–Mg Alloy Anodes on LLZO Type Solid Electrolytes as a Function of Temperature and Pressure. *Adv Energy Mater* **2019**, *9* (44). <https://doi.org/10.1002/aenm.201902568>.
- (24) Fuchs, T.; Haslam, C. G.; Richter, F. H.; Sakamoto, J.; Janek, J. Evaluating the Use of Critical Current Density Tests of Symmetric Lithium Transference Cells with Solid Electrolytes. *Adv Energy Mater* **2023**, *13* (45). <https://doi.org/10.1002/aenm.202302383>.